



# Impacts of rainfall features and antecedent soil moisture on occurrence of preferential flow: A study at hillslopes using high-frequency monitoring

Zhenyang Peng, Hongchang Hu, Fuqiang Tian, Qiang Tie, Sihan Zhao

Department of Hydraulic Engineering, State Key Laboratory of Hydroscience and Engineering, Tsinghua University, Beijing 100084, China

*Correspondence to*: Fuqiang Tian (tianfq@tsinghua.edu.cn)

**Abstract.** In order to evaluate influences of rainfall features and antecedent soil moisture on occurrence of preferential flow, observation was conducted at 12 sites within a 7-km$^2$ catchment, by applying the high-frequency monitoring approach. Totally

65 rainfall events were selected to compare among sites, and preferential flow was inferred when (i) responses of soil moisture did not follow a linear sequence with depth, or (ii) penetration velocity of wetting front in at least one horizon exceeded the threshold, which was set to be 5-10 times of the saturated hydraulic conductivity of soil matrix at different depths. Results showed that frequency of preferential flow was 40.7% in average, but varied from 17.9% to 74.3% among the sites. Correlations between the frequency and rainfall features, i.e. rainfall amount, duration, maximum and average intensity, were

well fitted by logarithmic curves. Rainfall amount, which was most prominently correlated with frequency ($R^2$=0.93), was regarded as the dominant driving factor of preferential flow, while average intensity was in second ($R^2$=0.90). Antecedent soil moisture was also significantly correlated with the frequency. However, this should largely be attributed to the differences of soil moisture among sites, since varying range of soil moisture at a specific site was not wide enough to influence the frequency significantly. Further examination suggested that topography and surface cover (dead leaves and humus) were the controlling

factors of both infiltration amount and occurrence of preferential flow, as water was more readily to infiltrate into soils and preferential flow was more readily to occur when slope gradient was small and surface cover was thick, while soil moisture was more likely to be a consequence of water storage capacity, rather than an inducer of preferential flow. This knowledge could be helpful in understanding the partitioning of surface runoff and infiltration, as well as runoff processes in catchments with complex topography and underlying conditions.

## 25  1. Introduction

Preferential flow refers to all kinds of phenomena where water moves along preferred pathways through the soil profile allowing water to bypass part of the soil matrix (Hardie et al., 2011), and has long been recognized to be ubiquitous, widespread and relatively densely distributed in soils (Flury et al., 1994; Jarvis, 2007). Given its fast velocity, preferential flow would allow water to move to greater depths, at faster rates, than predicated by the Richards equation (Hendrickx and Flury, 2001).



It may as a result, dominate subsurface storm runoff (Vogel et al., 2010) as well as groundwater supplement (Guo, 2008; Mathieu and Bariac, 1996; Shurbaji and Campbell, 1997), and eventually influence the runoff of a catchment (Köhne et al., 2009; Lin and Zhou, 2008; Tromp-van Meerveld and McDonnell, 2006).

Among the many (Jarvis, 2007; Lin, 2006; Niu, 2003), rainfall features and antecedent soil moisture are two essential control factors of occurrence and extent of preferential flow. Basically, preferential flow is more likely to occur when rainfall amount is large, duration is long and intensity is high (Guo, 2008; He et al., 2005; Heppell et al., 2002). And Heppell et al. (2002) even categorized rainfall induced preferential flow into two types: intensity driven and duration driven. Moreover, Cheng et al. (2006) reported a 26mm threshold of 24-hour rainfall amount for preferential flow to significantly influence runoff in Three-Gorge Region of China and suggested that there would be a chance for preferential flow to occur at this location when maximum rainfall intensity reached 0.075mm/min. McGrath et al. (2008) also implied the impact of rainfall intensity on preferential flow rates, and later proposed a 19-mm threshold of rainfall amount for preferential flow to occur, since they did not observe significant leaching of herbicide when rainfall amount was lower than this threshold (McGrath et al., 2010). However, column experiment of undisturbed soils conducted by Wu et al. (2014) indicated that the growth of preferential flow rates along with increasing rainfall intensity would be limited by saturated hydraulic conductivity. And in sharper contrast, Hardie et al. (2013) argued that rainfall features (i.e. rainfall amount, intensity, duration) are of minimal importance on occurrence of preferential flow when soil is initially dry, since water repellence would diminish the advantage of heavy rain's infiltration momentum and prevent ponded water entering the soil readily.

Though Hardie et al.(2013) suggested antecedent soil moisture to be a prominent control factor of preferential flow occurrence, researchers have not reached agreement on whether preferential flow is more likely to happen in drier or in wetter soils until recently (Merdun et al., 2008). On one hand, size of soil cracks would generally decrease as soil became progressively wetter (Greco, 2002; Wang et al., 2004), and wetting front would be unstable as water infiltrated into water repellent soils (Dekker and Ritsema, 2000), thus water would infiltrate faster (Lin et al., 1998; Hardie et al., 2011) and penetrate deeper (Hardie et al., 2013) when soil is initially drier, while preferential flow is more likely to occur (Merdun et al., 2008) and contribute more on water and solute movements (Shipitalo and Edwards, 1996). However, since larger pores, the major pathways of preferential flow (Granovsky, et al., 1994), is assumed to be hydraulic active when soil is wetter (Jaynes et al, 2001; Kung et al., 2000), it is also supported by many studies that higher antecedent soil moisture would increase the depth to which preferential flow penetrates as well as increasing total percolate volume (e.g. Jarvis, 2007; Sheng et al, 2012).

Among the numerous approaches (Allaire et al., 2009), methods based on interpreting information of soil moisture distribution and its movement have been widely applied in quantifying preferential flow (e.g. Ellsworth and Boast, 1996, Cambouris et al., 2006). Lin and Zhou (2008) inferred evidence of preferential flow during rainfall when soil moisture responses did not follow a linear sequence with depth, and this criterion was later applied and improved by other researchers. For example, Fu et al. (2011) applied this criterion combined with the hysteresis time, which is the time gap between the start of surface runoff and water content to reach stable. Hardie et al. (2013) categorized infiltration into 5 types, and assumed that





besides Lin and Zhou's (2008) criterion, high penetration velocity of wetting front, which was arbitrarily set to be 10 times higher than of saturated hydraulic conductivity of soil matrix, was also an indicator of preferential flow.

In this study, we sought to quantify the presence of preferential flow at forestry hillslopes, by interpreting variations of soil moisture which were obtained by high frequency monitoring, and to evaluate the impact of rainfall features and antecedent soil moisture on occurrence of preferential flow at different locations on slopes.

## 2. Methodology

### 2.1 Experiment site

Observations were carried out at Xitaizi experimental catchment (116°37′E, 40°32′N), which is located in the north part of Beijing, China, with an area around 7km² and altitudes ranging between 600m and 1200m (Fig.1). Local mean annual rainfall is 625mm (1989-2013) and more than 80% of the rainfall amount is precipitated between June and September, showing an uneven distribution across the year. Local mean annual temperature is 11.45℃ (1989-2013), and soils are frozen from late November to middle March, when daily mean air temperature is continuously below 0℃. Hillslope at this catchment is well covered by forest, which is mainly composed by Larix gmelinii and Populus davidiana, and the canopy coverage could reach 98% in summer (Source: SPOT, August 2013) when leaves were exuberant.

### 2.2 Observations of rainfall and soil moisture

Totally 12 sites were selected for observations in this catchment (Fig.1), of which 4 sites (FH1-FH4) are located at relatively flat areas with no canopy cover within at least 5 meters. The other 8 sites were distributed along two canopy-covered hillslopes, from high to low.

At each site, TDR technique based CS616 probes (Campbell Scientific, Inc., Logan, USA) were used to measure the volume water content in soils. They were hierarchically buried within the top 60cm (FH1-FH4), 80cm (S1H1-S1H5) or 90cm (S2H1-S2H3) of soils, and depth increments between adjacent probes were all 10cm. Meanwhile, 9 TE525 tipping bucket rain gauges (Texas Electronics, Inc., Dallas, USA) were set on the ground near some of the sites, so as to monitor rainfalls at different locations.

Both soil moisture content and rainfall amount were continuously measured in every 10 minutes, and the measuring results were automatically recorded by CR1000 data loggers (Campbell Scientific, Inc., Logan, USA). The monitoring period of FH1-FH4 was from August 2013 to October 2015, while that of the other 8 sites were from August 2014 to October 2015. Detailed information of monitoring settings was listed in Table 1.





### 2.3 Soil properties

Soils at the hillslopes are not well developed. They are commonly mixed with gravels, sizes of which vary from millimeters to decimeters. The bedrocks are shallowly buried and occasionally exposed, leading to small soil thicknesses, which presumably range between 0 and 2m.

Eighty four groups of soil samples were collected from the profiles at all sites except FH4. Content of organic matter was measured by titration method, and particle sizes were measured by Mastersizer 2000 (Malvern Instruments Ltd., Malvern, UK). Saturated hydraulic conductivities of soil matrix were estimated based on particle composition of each sample, by employing the hierarchical artificial neural network model of the pedotransfer package ROSETTA Lite V1.1 (Schaap et al., 2001), which was widely applied for estimating hydraulic parameters of soils among studies (e.g. Nelson and Rittenour, 2015; Rieckh et al.,
2012).

Data showed that soil compositions varied notably among the sites. Meanwhile, contents of organic matter, clay and silt generally decreased with depth, leading to higher saturated hydraulic conductivity. Detailed information at each depth was not listed in Table 2, but was covered by the ranges.

### 2.4 Determination of rainfall events

Only rainfalls during unfrozen seasons (April to October) were taken into account. Also, since preferential flow in shallow soils would normally occur within hours after rainfall, rainfall events were divides into ones with relatively short durations in this study, which were normally within 24 hours, and not more than 48 hours for continuous heavy rains. Starting and ending of a rainfall event was determined according to the following rules.

An event was assumed to start when (i) water content at all monitored depths did not change significantly in the previous
1 hour, or more specifically speaking, standard deviation of water content should be less than $0.001cm^3/cm^3$; and (ii) rainfall amount in the next 1 hour should be no less than 1mm. An event was assumed to end if rainfall amount in the next 6 hours was less than 1mm.

### 2.5 Inference of preferential flow

Hardie's et al. (2013) method was applied to infer preferential flow in this study. Given accuracies of the applied TDR
probes were $0.025cm^3/cm^3$ as was suggested by the producer, and were normally between $0.01-0.03cm^3/cm^3$ as was calibrated by many authors (e.g. Fan et al., 2015; Zanetti et al., 2015), a $0.002\ cm^3/cm^3$ threshold was set to quantify the responses of water content to infiltration according to a bunch of studies (e.g. Blume et al., 2009; Lin and Zhou, 2008). Variations higher than the threshold were regarded as evidence of response, while variations lower than the threshold were assumed to be noises of instruments.

In the next step, penetration velocities of wetting front were estimated by eq. (1).



$$v_{wf} = \frac{\Delta h}{\Delta t}$$

(1)

where, $v_{wf}$ was penetration velocity of wetting front, cm/h; $\Delta h$ was soil thickness between adjacent TDR probes, which was

10cm in this study; $\Delta t$ was the time gap between responses of two adjacent probes, if they responded in sequential with depth.

Based on results of above calculations, infiltrations could be classified into 5 types, and 2 types were signs of preferential

flow (Hardie et al., 2013):

(i) PF-ns: occurred when the responses of probes did not follow a linear sequence with depth;

(ii) PF-rate: occurred when velocities of wetting front were high enough in at least one layer, e.g. Hardie et al.( 2013) set
the threshold to be about 10 times of saturated hydraulic conductivity of soil matrix. However, Hardie's et al.( 2013) arbitrary
setting may have been too conservative, and thus lead to omissions of preferential flow. Indeed, when water moved as uniform
flow, velocity of wetting front may exceed saturated hydraulic conductivity by folds at the beginning of infiltration. However,
this high exceedance would be limited within a short time and within a shallow depth, which was normally several centimeters.
It would then decrease almost exponentially as saturated layer getting thicker, and lead to approximately piston flow during
continuous infiltration (Peng et al., 2012). Nevertheless, considering the errors of estimated hydraulic conductivity, threshold
was also set to be 10 times of saturated hydraulic conductivity in the top 10cm in this study, while it was set to be 5 times in
the layers below, since the exceedance of penetration velocity to saturated hydraulic conductivity was different with depths.
Meanwhile, the maximum saturated hydraulic conductivity among FH1-FH3 at a certain depth was arbitrarily regarded as the
value of this depth at FH4, as no data on soil properties at this site was obtained.

## 3. Results

### 3.1 Rainfalls

The whole monitoring period was divided into two parts, by the date 2014/8/22, when rain gauges on slopes all started to work.
Rainfall amount varied notably from one site to another, as was showed in Table 3. Rainfall amounts observed at FH3 and
FH4 were notably higher than at other sites in both periods, while amounts observed at FH2 were always the lowest among
the sites that were located at flats. Amounts observed at Slope I were relatively spatially homogeneous, though the maximum
difference among the 4 rain gauges could be as much as 46.8mm. More rainfall events were obtained generally at sites with
larger observed rainfall amount. Totally 93 events were obtained at FH3 and FH4, while only 43 events were obtained at S2H1.

The spatial variations discussed above were presumably because of two reasons. One could be ascribed to the occasional
dysfunction of rain gauges. For instance, the logger failed to record the data at S2H1 from 2014/10/14 to 2014/10/31, during
which rainfall occurred. Meanwhile, leaves, sprigs, gravels, mud, etc., that fell onto the surface of rain gauges occasionally,
may also lead to errors of monitoring. Spatial heterogeneity of rainfall could be the other reason. Data showed that in a given
time range, rainfall amount and its temporal distribution could hardly be same at different sites. Rainfall may be temporally



concentrated at one location, while at another location, it may be too sparsely distributed or the observed rainfall amount may be too low, to be regarded as an event according to the criterion described above (Fig. 2).

In order to compare between different sites, rainfall events observed by all rain gauges at the same time were selected. It was totally 26 events in period I and 39 events in period II, respectively. Fig 3 showed that frequency distributions of rainfall amounts were rather close to each other among the sites located at flats, that they could basically be well fitted by a Pearson-III curve with uniform parameters. And so were the frequency distributions of sites located at slopes.

However, intuitive comparison between the two diagrams may lead to conclusion that frequency distributions at flats and slopes were different with each other, given skewness of the later (6.99) was over two times larger than that of the former (2.62), and resulted in steeper decline of rainfall amount when frequency was relatively low.

Since none of the rainfall features, i.e. rainfall amount, duration, maximum intensity and average intensity, was likely to follow a Gaussian distribution, the Kolmogorov-Smirnov test was applied in further analysis to quantify the differences between every two sites as well as between each site and Gaussian distribution. Significant difference between every two distributions was set as the null hypothesis, and the significance level was set to be 0.05. Results were displayed in Fig.4, in which the values of self-comparisons were set to be blank.

Significances (P-values) between sites and Gaussian distribution were all above 0.05 on all rainfall features (Fig.4). They were normally over 0.1 and could be larger than 0.5 at some occasions (e.g. FH1 vs Gaussian on rainfall amount), indicating that the null hypothesis should be accepted, and the above assumption that none of the frequency distributions on all rainfall features at all sites followed a Gaussian distribution was testified.

In contrast, significances of differences between every two sites were much lower. They were normally less than 0.02, and almost all of them were less than 0.05, except 3 groups of comparisons, i.e. FH1 vs FH4 and FH4 vs S1H1 on rainfall amount, FH3 vs S1H2 on average intensity. Generally, the null hypothesis should be rejected when comparing between two sites, indicating that the selected rainfall events were not spatially varied, at least not significantly. As such, we assumed that rainfall events were comparable among all sites, and observed data of a rain gauge could be applied to those sites without rain gauges and located nearby.

**3.2 Occurring Frequency of preferential flow**

Fig.5 (a) is an exemplary pattern of PF-ns, in which increments of soil moisture at 30cm and 40cm exceeded the threshold (0.002 $cm^3/cm^3$) earlier than at 10cm and 20cm, and indicated occurrence of bypass flow within the top 30cm of soils. In Fig.5 (b), soil moisture responded to rainfall in linear sequence with depth. Probe at 10cm responded within 30min since rainfall started, and times gaps of responses in the two layers below were less than 20min and less than 10min, respectively. This meant that penetration velocities of wetting front in the top 30cm were no less than 20cm/h, and could be over 60cm/h, which were all far faster than the estimated saturated hydraulic conductivity of soil matrix at this site (see Table 2), and inferred occurrence of PF-rate.



As a matter of fact, PF-ns may also be accompanied with fast penetration of wetting front. Taking Fig.5 (a) as an example, besides the non-sequential responses, penetration velocity of wetting front in soils between 10-20cm was about 60cm/h, which also inferred occurrence of preferential flow. Nevertheless, soil profile was regarded as a whole to quantify the occurrence of preferential flow in this study, and only the patterns in which probes responded quickly and sequentially, were regarded as PF-rate in following discussions.

Results showed that preferential flow occurred at all sites (Fig. 6), though not always at the same time. The average frequency of preferential flow was 40.7%, if PF-ns(28.8%) and PF-rate (11.9%) were combined together. Number of sites at which preferential flow occurred during an event, varied notably among the events. Preferential flow did not occur at any of the sites during some events in both periods, while it occurred at all the 4 sites during several events in period I. During most of the events, especially in period II, preferential flow was more likely to occur at a bunch of, rather than at none or at all of the observed sites, and the number of sites that had preferential flow occurred at the same time could be up to 9, i.e. during the 23th and 28th event in period II.

Meanwhile, frequencies of preferential flow also varied notably from one site to another. The frequency could be lower than 20% at S1H3 and S1H4 during period II, and it could also be higher than 70% at FH3 during the same period. This hinted possible influences of underlying surface conditions on occurrence of preferential flow, since rainfall features were statistically same at all sites.

Whilst Jarvis (2007) argued that preferential pathways were dynamic since the soils were exposed to temporal variations in climate, biological activities and artificial managements, no significant difference on frequencies of preferential flow was observed between the two periods. For instance, frequencies were 54% at FH4 in both periods, while maximum interannual difference at FH1-FH4 was 7.6%. This result implied a temporal stability of the underlying surface conditions in this catchment, since rainfall features were close in the two periods (P<0.05). This stability may be attribute to the relatively short observing period on one hand, while on the other hand, minimal human intervenes were imposed on the 4 sites during observations, and thus no abrupt or distinguishable changes occurred. Though we are not sure how long would it take for the soil morphology to change significantly enough to influence the occurrence of preferential flow, one should be confident that data obtained at the same location within several years are comparable.

## 4. Discussion

### 4.1 On influence of rainfall features

Fig.7 showed that ranges and distributions of the rainfall feature values were not the same when different types of flow occurred. Nonetheless, those ranges were not so distinguishable among different types flow, that one may conclude that rainfall features were of minimal importance in determining flow types. Indeed, median values of each rainfall feature were very close to each other when PF-ns or PF-rate occurred, except on rainfall durations. But one should note that the durations covered or uncovered by the boxes still largely agreed with each other between PF-ns and PF-rate.



Whereas differences between the ranges of preferential flow occurred and non-occurred scenarios were more significant, than between the two types of preferential flow. Median values and ranges covered by boxes were notably higher when preferential flow occurred, and basically implied that preferential flow was more likely to occur when rainfall amount was large, duration was long, maximum intensity as well as average intensity was high.

The value ranges were then divided into 15 sub-ranges on each rainfall feature with non-uniform increments, and occurring frequencies of preferential flow of each sub-range were calculated. As Fig.8 showed, higher frequencies were generally obtained at larger values of all the 4 rainfall features, and full percent was reached when rainfall, maximum intensity and average intensity exceeded 100mm, 40mm/h and 25mm/h, respectively. However, these correlations were still largely uncertain, especially in the scenarios of duration and maximum intensity. For example, the frequency was only 30% when

duration was 20-25h, which was less than half of that when duration was 15-20h. Similar phenomenon also happened in the scenario of maximum intensity. Frequencies increase significantly along with maximum intensity in the range of 0-15mm/h, however, there was a sharp decline between the 10-15mm/h and 15-20mm/h, and it then increased again as the maximum intensity growth continuously, indicating possibility of sectionalized correlations between frequencies and maximum intensities.

Heppell et al. (2002) suggested that duration and intensity of rainfall were the two major driven factors of preferential flow. However in this study, correlation between frequency and duration was of least significance among the 4 features when fitted with logarithmic equation. The value of $R^2$ in this scenario was only about 0.39 and $R^2$ of linear correlation between the two was 0.47, which was not much higher. Nevertheless, correlation between average intensity and frequency was of $R^2$ higher than 0.90, indicating itself as a prominent driven factor of preferential flow, and agreed with Heppell's et al. (2002) argument.

Meanwhile, correlation between rainfall amount and frequency showed even higher significance ($R^2=0.93$), and thus rainfall amount should be regarded as the most prominent independent driven factor, since it was not significantly correlated with other rainfall features when preferential flow occurred ($R^2<0.02$, n=233).

## 4.2 On influence of antecedent soil moisture

Similar approaches were applied to analyze the correlation between antecedent soil moisture and occurrence of preferential

flow. The median value as well as the up limit value covered by the box that generated PF-ns was the highest among the three, while those with non-preferential flow occurred scenarios were the lowest (Fig.9). Further t-tests also indicated that antecedent soil moisture was significantly higher to generate preferential flow than to generate non-preferential flow, but those were not significantly different between PF-ns and PF-rate. Fig.10 showed more clearly that higher antecedent soil moisture generated preferential flow more frequently, and this correlation was also well fitted by the logarithmic curve with value of $R^2$ over 0.80.

However, above conclusions may have been arbitrary when variations among the sites were taken into account. As was showed in Fig.11, frequencies of preferential flow varied significantly from one site to another, and so were antecedent soil moisture. Generally, preferential flow occurred more frequently at sites with higher antecedent soil moisture ($R^2=0.81$), which seems to be a further validation of above conclusion.





On the contrary, one should note that soil moisture variations during period II, which ranged between 0.02 to 0.04 cm$^3$/cm$^3$, were almost consistent among the sites, and were relatively small. This indicated that soil moisture at some sites were continuously higher than at others all through the period. As such, high frequencies in Fig. 10 were largely contributed by sites that with continuously higher soil moisture, e.g. FH3 and FH4, while the low frequencies were mainly encumbered by the sites with continuously low soil moisture, e.g. the sites on Slope I.

Meanwhile, individual analysis of each sites showed that frequency was not significantly correlated with antecedent soil moisture, presumably because the variation of soil moisture at a specific site was too small to significantly influence the occurrence of preferential flow. Therefore, though we cannot deny the possible influence of antecedent soil moisture on occurrence of preferential flow, we should admit that it may not be the unique or direct reason when explaining the variation of the frequency among sites.

Beside rainfall features and antecedent soil moisture, Lin and Zhou (2008) listed a bunch of other factors, such as hillslope position, slope orientation and gradient, the underlying bedrock fracture and orientation, or some forms of combinations, which may also influence the occurrence and extent of preferential flow at hillslopes. While in this study, soil moisture was generally lower and preferential flow less frequently occurred in sites located at slopes than at flats. And this could basically be ascribed to two reasons. One is slope gradient, since water fell onto slopes would be more readily to generate surface runoff, rather than to be ponded and infiltrate into soils. The other is surface cover, which was composed by dead leaves and branches, as well as humus. It has been widely accepted that the covering layer played an essential role in water storage at hillslopes. Amount of water stored in the covering layer could be up to 5 times of the dry weight (Zhao et al., 2002). It would consequently delay the generation and diminish the amount of surface runoff, and promote infiltration (Wu et al., 1995; Zhang et al., 2000). Moreover, studies showed that decaying of the covering layer and intensified biological activities would induce the occurrence of preferential flow, and may lead to subsurface dominated runoff in a catchment (Kim et al., 2014; Wang et al., 2004).

For example, in Fig.12, soils at FH3 responded much stronger than at S1H3 during an event. Rough estimation by assuming soil was linearly wetted in layers showed that maximum increment of water storage at S1H3 (1.7mm) was minimal when compared with rainfall amount (8.9mm), indicating most of the water had escaped from the slope through surface runoff. Additional observation at the foot of Slope I also showed that undetectable amount of water exuded from the subsurface layers, while the amount of surface runoff was prominent. During the same period, increment of water storage at FH3 (11.8mm) was significant, and was even larger than the rainfall amount (9.6mm), indicating the site had been ponded with water, and may have received surface recharges from upslope areas. As Jarvis (2007) concluded that though ponding was not necessary, water pressure should reach close to saturation at least around the macropores to generate non-equilibrium flow, it was not surprising that preferential flow occurred at FH3, but not occurred at S1H3.

Since (i) Slope I was steeper than Slope II, and FH1-FH4 were located at flat areas, and (ii) surface cover at Slope II was averagely thicker than at Slope I, where soils were sometimes exposed, difference between the FH3 and S1H3 could be explained by the two reasons discussed above, and so were the differences among rest sites. And it was reasonable to conclude that antecedent soil moisture could not be the direct reason of the variation on frequencies of preferential flow. On the contrary,



it was the topography and surface cover that determined the infiltration and the occurrence of preferential flow, while the soil moisture was merely a consequence of infiltration and water storage capacity at sites.

## 5. Conclusion

In this study, Preferential flow was inferred when (i) responses of soil moisture did not follow a linear sequence with depth (PF-ns), or (ii) penetration velocity of wetting front in at least one horizon exceeded the threshold (PF-rate). It was concluded that preferential flow occurred at all the observed 12 sites in a semi-humid head water catchment, North China. The occurring frequency of preferential flow was averagely 40.7%, but varied notably among the sites, which ranged from 17.9% to 74.3%. Preferential flow was more likely to occur in the form of PF-ns than in the form of PF-rate, with frequencies of each were 28.8% and 11.9%, respectively, though PF-ns may also be accompanied with high penetrating velocity of wetting front.

Frequency of preferential flow showed significantly positive correlations with all the 4 rainfall features, i.e. rainfall amount, duration, maximum and average intensity. The correlations were well fitted by logarithmic curves, and the $R^2$ (0.93) between frequency and rainfall amount was the highest, indicating it was the dominant driven factor, while average intensity was in second ($R^2=0.90$).

Correlation between antecedent soil moisture and frequency was also significant, and was well fitted by logarithmic curves. However, differences of antecedent soil moisture was mainly contributed by the variations among the sites, while at a specific site, variation of soil moisture during the observing period was not large enough to show a significant correlation with frequency. It was thus concluded that though we cannot deny the possible influence of antecedent soil moisture on occurrence of preferential flow, topography and surface cover were more likely to be the essential controlling factors. Water was more readily to infiltrate into soils and preferential flow was more readily to occur when slope was less steep and covering layer was thicker, therefore the soil moisture was more likely to be a consequence of water storage capacity, rather than a inducer of preferential flow.

## Acknowledgemetns

This research was funded by the National Science Foundation of China (51190092) and the Key Lab (2016-KY-03).

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





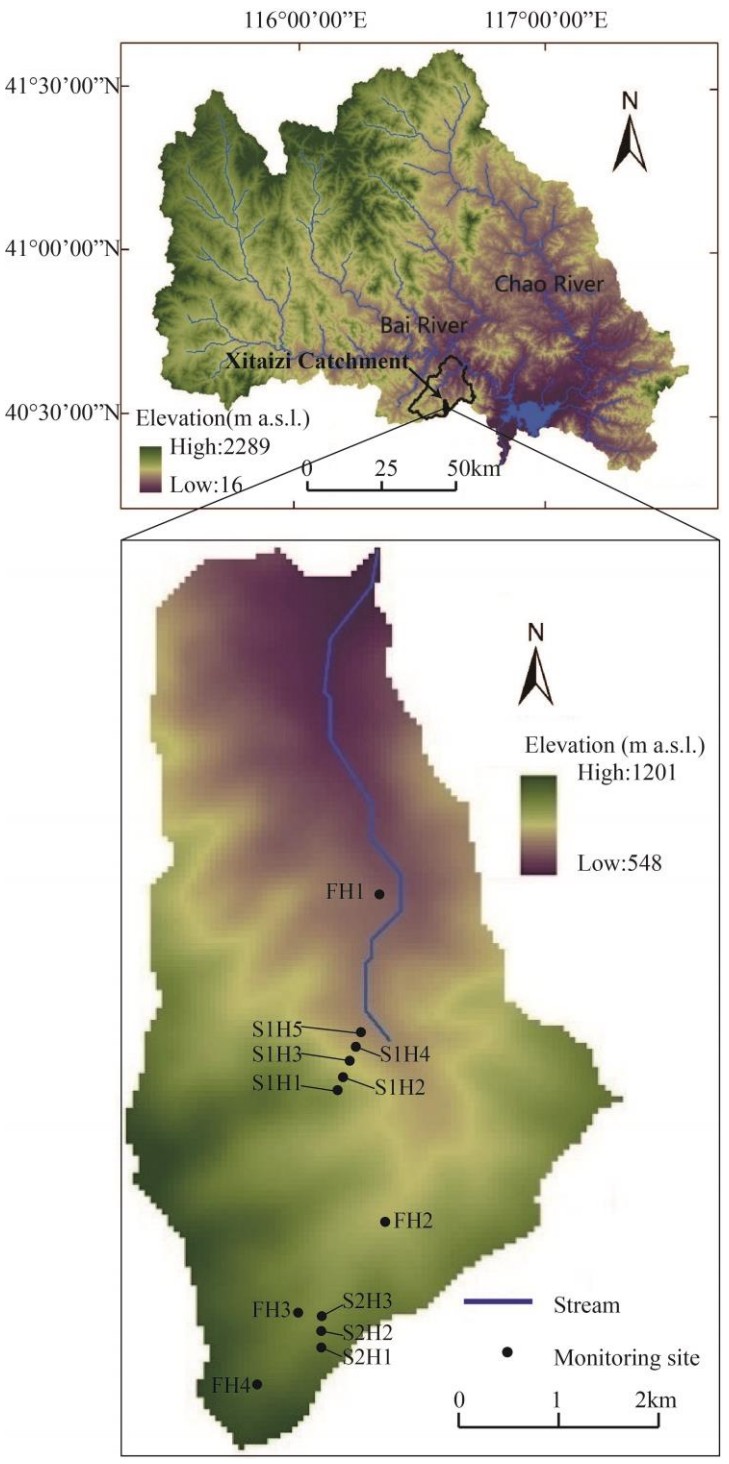

**Fig.1 Contour maps of Xitaizi Catchment and locations of observation sites**




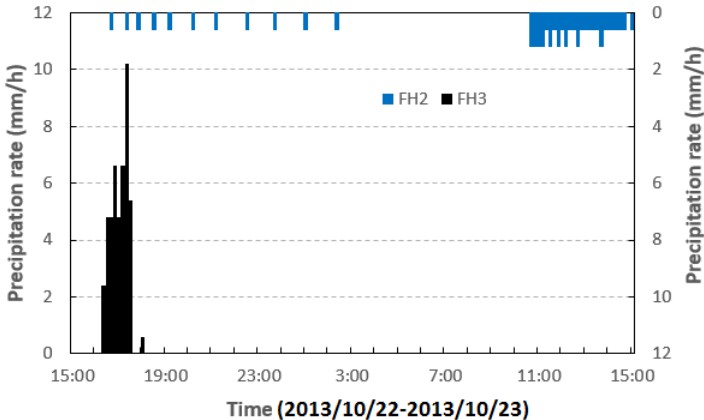

**Fig 2. Comparison of rainfalls observed at different locations during the same period**

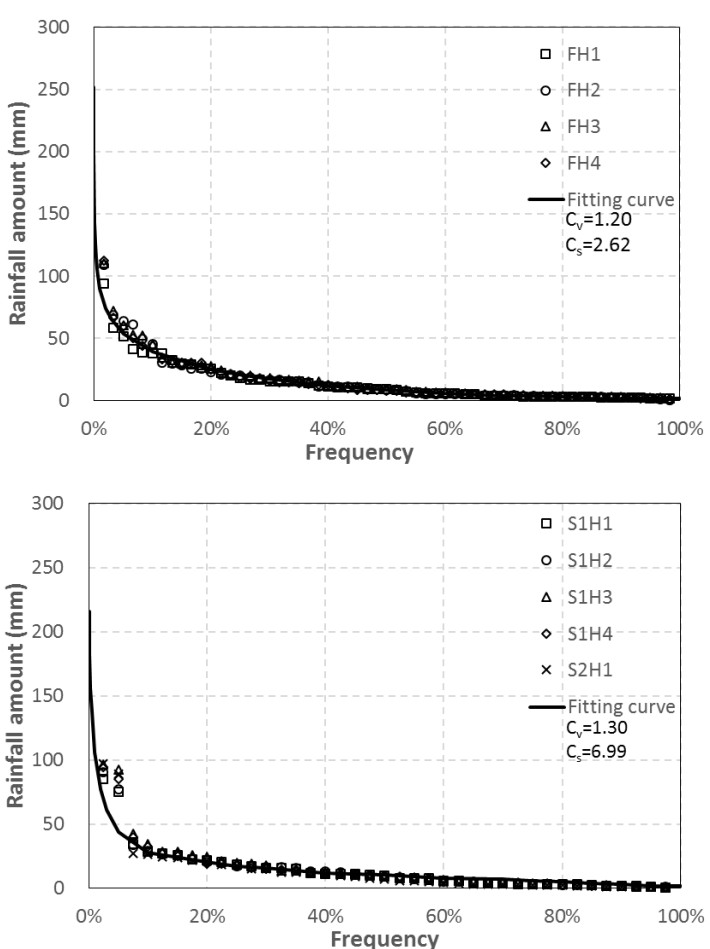

5     **Fig.3 Frenqucy distributions of rainfall amount at different sites deduced by selceted events**



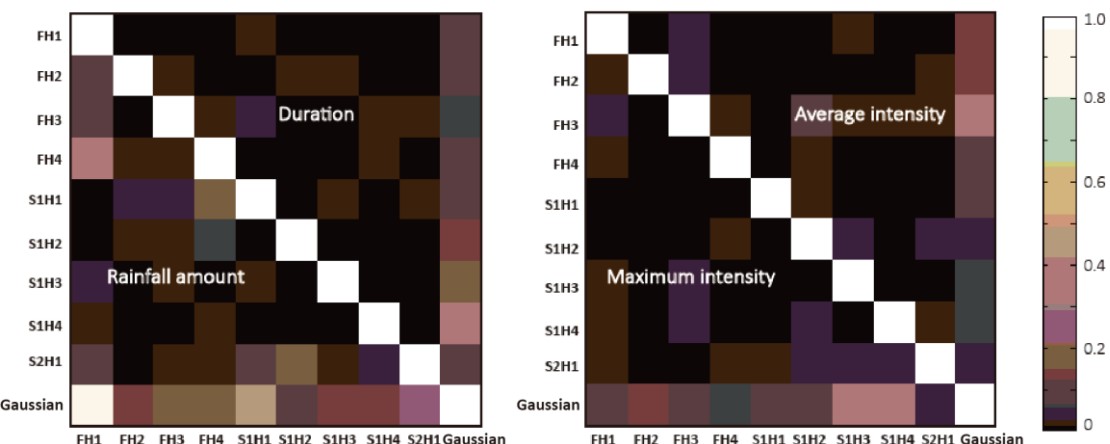

**Fig.4 Significances (P-values) of differences between sites on rainfall features**

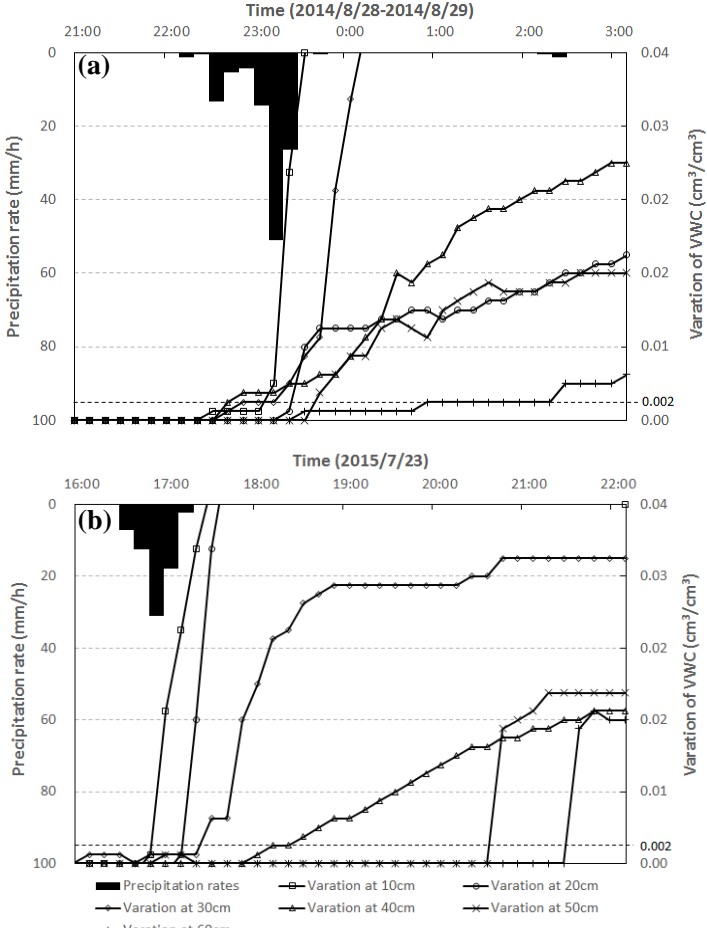

**Fig. 5 Examples of the two types of preferential flow inferrings at FH 3: (a) PF-ns and (b) PF-rate**



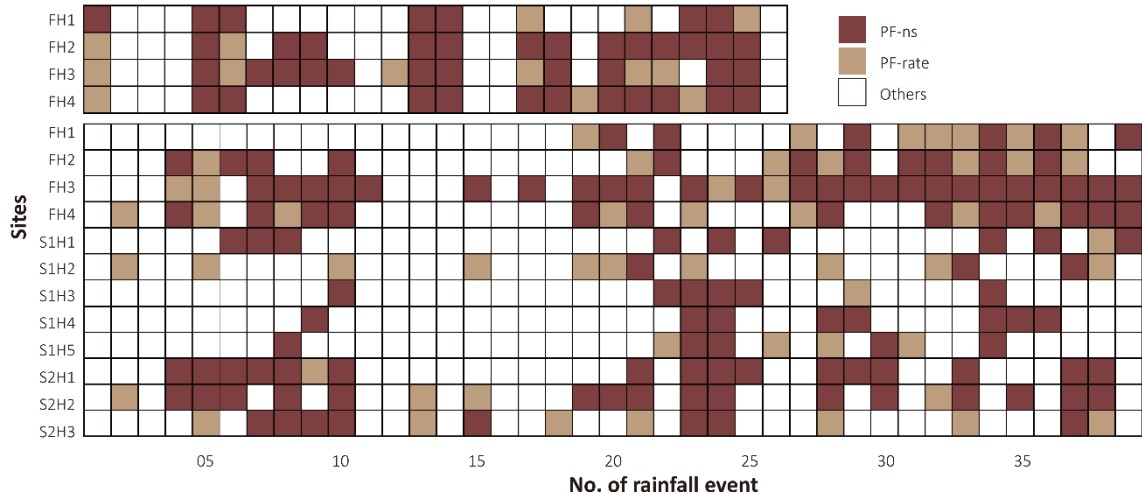

**Fig. 6 Occurrences of preferential flow at sites during peroid I (above )and period II (below)**

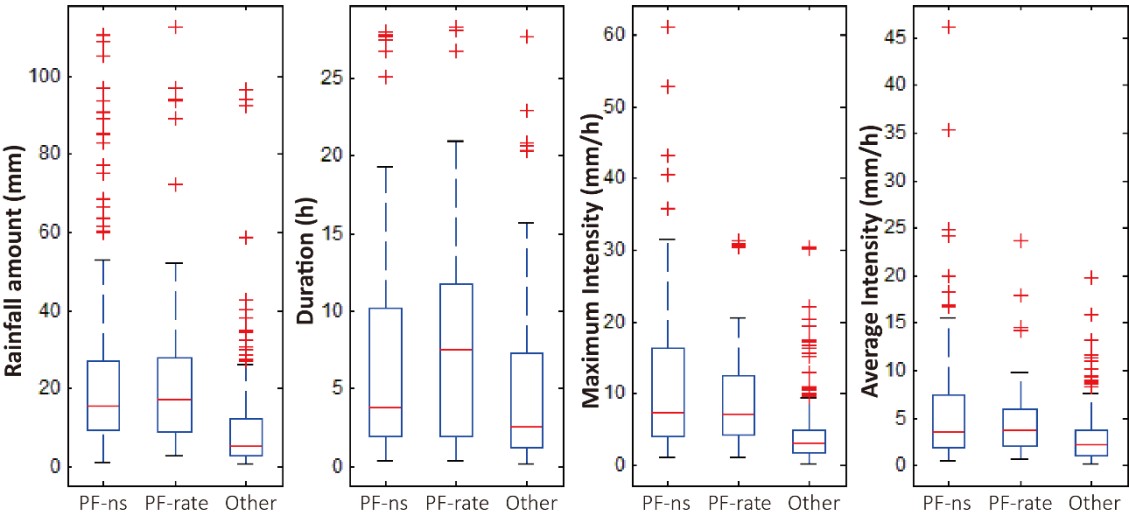

**Fig. 7 Value distributions of rainfall features when different types of soil water flow occurred**





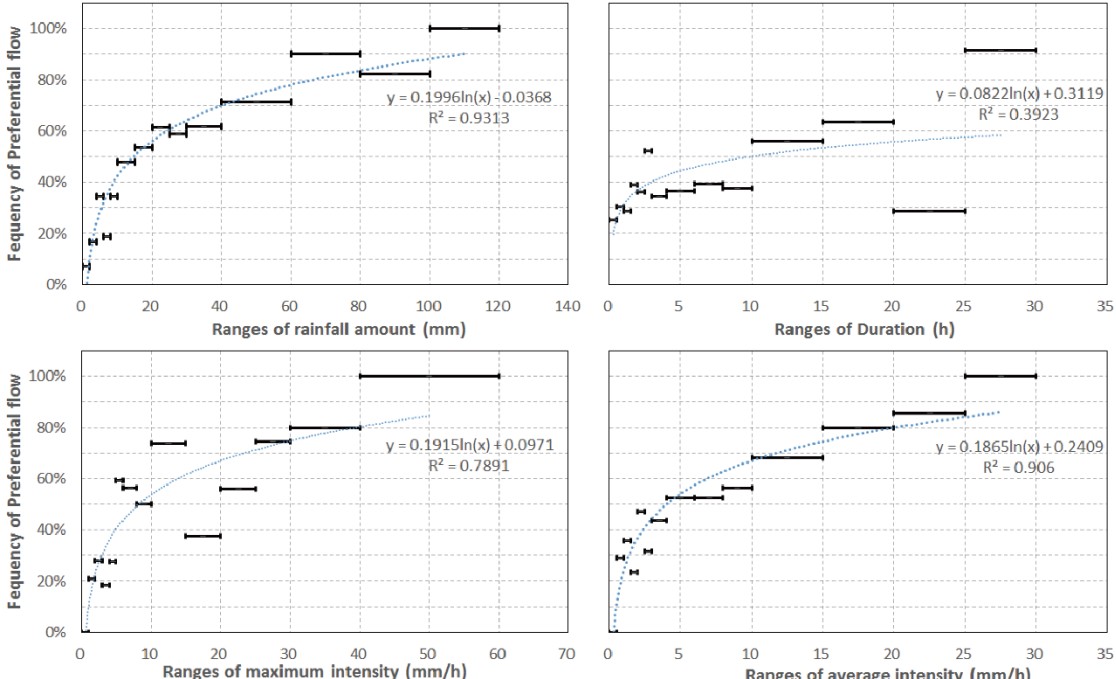

**Fig. 8 Relationship between values of rainfall features and frequency of preferential flow**





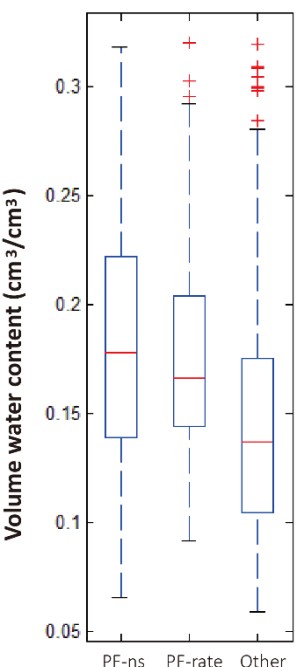

**Fig.9 Antecedent soil moisture when different types of flow occurred**

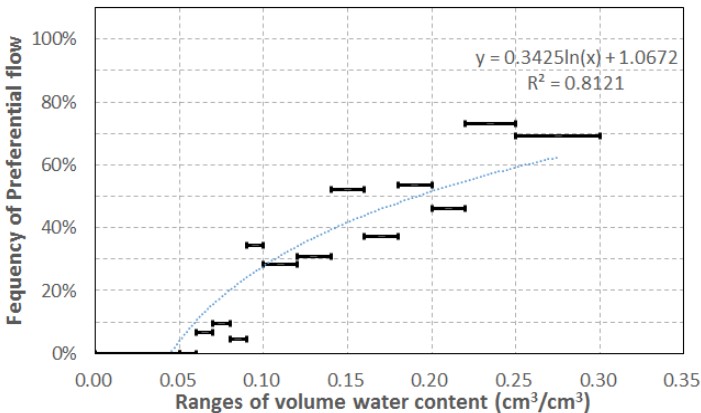

**Fig. 10 Relationship between average antecedent soil mosisture (0-60cm) and frequency of preferential flow**





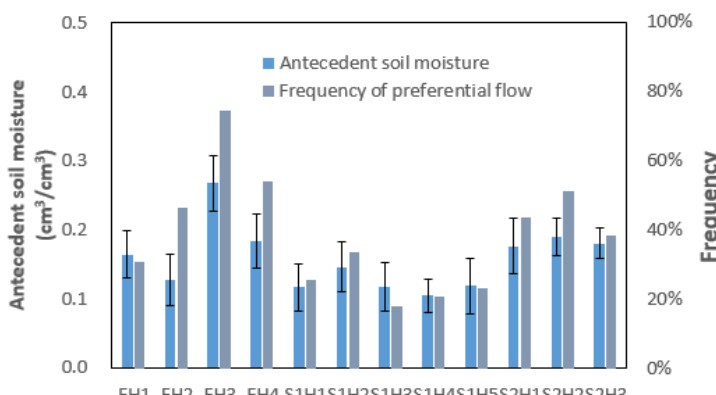

**Fig 11 Average antecedent soil moisture (0-60cm) and frequencies of preferential flow at all sites in period II**

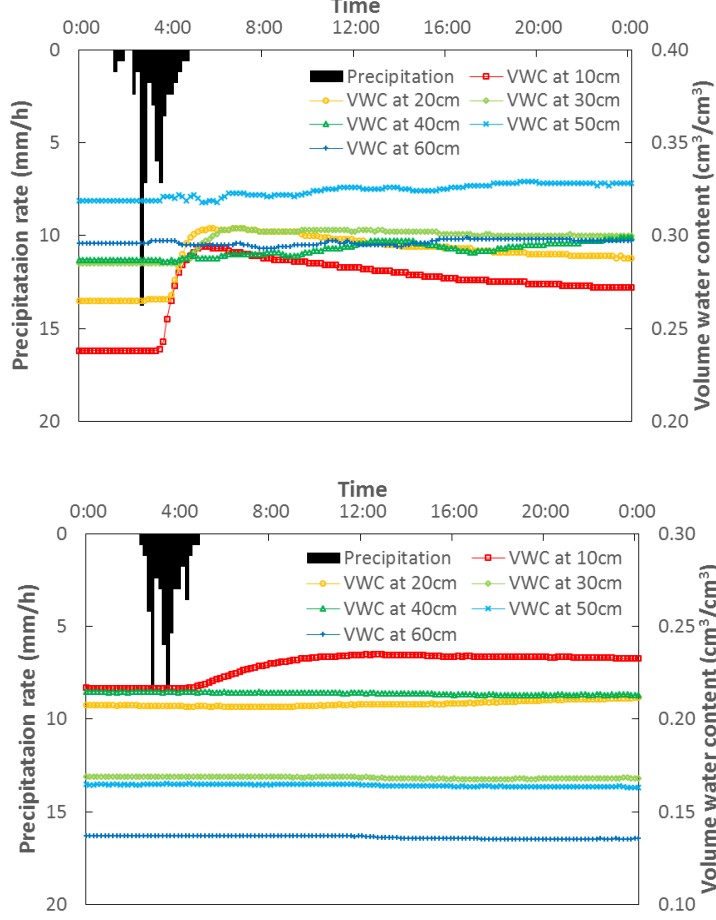

5 **Fig 12 Responses of volume water content (VWC) to rainfall at FH3 (above )and S1H3 (below) on 2015/9/18**



**Table 1 Monitoring settings at each site**

| Name of Sites | Locations | With or without canopy cover | Depths of TDRs (cm) | With or without rain gauge | Monitoring period |
|---|---|---|---|---|---|
| FH1 | Flat | without | 10-60 | with | 08/2013-10/2015 |
| FH2 | Flat | without | 10-60 | with | 08/2013-10/2015 |
| FH3 | Flat | without | 10-60 | with | 08/2013-10/2015 |
| FH4 | Flat | without | 10-60 | with | 08/2013-10/2015 |
| S1H1 | Slope1 | with | 10-80 | with | 08/2014-10/2015 |
| S1H2 | Slope1 | with | 10-80 | with | 08/2014-10/2015 |
| S1H3 | Slope1 | with | 10-80 | with | 08/2014-10/2015 |
| S1H4 | Slope1 | with | 10-80 | with | 08/2014-10/2015 |
| S1H5 | Slope1 | with | 10-80 | without | 08/2014-10/2015 |
| S2H1 | Slope2 | with | 10-90 | with | 08/2014-10/2015 |
| S2H2 | Slope2 | with | 10-90 | without | 08/2014-10/2015 |
| S2H3 | Slope2 | with | 10-90 | without | 08/2014-10/2015 |

**Table 2. Composition and hydraulic properties of soils at each site**

| Name of Sites | Organic Matter content (%) | Particle sizes: proportion by weight (%) | | | Hydraulic conductivity (cm/h) |
|---|---|---|---|---|---|
| | | 0-0.002mm | 0.002-0.05mm | 0.05-2mm | |
| FH1 | 1.4-9.8 | 0.6-3.5 | 15.1-47.6 | 49.5-84.3 | 2.7-7.4 |
| FH2 | 1.4-5.0 | 0.3-3.1 | 15.0-39.8 | 57.4-84.8 | 2.6-8.0 |
| FH3 | 3.0-9.7 | 1.1-5.3 | 18.7-50.0 | 45.5-80.2 | 2.0-5.1 |
| S1H1 | 1.6-8.0 | 3.0-7.8 | 35.2-79.6 | 12.6-61.5 | 1.7-2.8 |
| S1H2 | 0.7-8.4 | 6.1-8.4 | 73.1-78.7 | 14.1-20.9 | 1.6-2.3 |
| S1H3 | 1.0-5.0 | 3.5-8.4 | 46.6-82.3 | 9.4-49.9 | 1.5-2.7 |
| S1H4 | 0.6-3.9 | 0.1-5.3 | 22.9-67.2 | 27.5-76.8 | 2.8-4.5 |
| S1H5 | 0.9-5.9 | 1.8-6.7 | 32.2-71.5 | 23.1-65.9 | 2.2-3.3 |
| S2H1 | 1.9-8.3 | 5.2-8.2 | 65.1-76.5 | 17.2-29.7 | 1.8-2.9 |
| S2H2 | 1.1-6.3 | 0.9-6.7 | 24.6-63.2 | 31.1-74.4 | 2.5-4.0 |
| S2H3 | 1.4-9.8 | 0.6-3.5 | 15.1-47.6 | 49.5-84.3 | 2.7-7.4 |





**Table 3 Rainfall amount and number of events observed at different sites**

| Name of Sites | Period I: 2013/8/8-2014/8/21 | | Period II: 2014/8/22-2015/10/31 | |
|---|---|---|---|---|
| | Rainfall amount (mm) | Number of events | rainfall amount(mm) | Number of events |
| FH1 | 445.2 | 31 | 778.4 | 51 |
| FH2 | 439.9 | 24 | 634.2 | 44 |
| FH3 | 559.6 | 41 | 866.4 | 52 |
| FH4 | 555.0 | 41 | 823.0 | 52 |
| S1H1 | | | 716.6 | 51 |
| S1H2 | | | 773.1 | 52 |
| S1H3 | | | 773.4 | 49 |
| S1H4 | | | 724.0 | 46 |
| S2H1 | | | 585.2 | 43 |