# Peer review of "Impacts of rainfall features and antecedent soil moisture on occurrence of preferential flow: A study at hillslopes using highfrequency monitoring"

_Hydrology and Earth System Sciences, 2016_

## Referee Comment (RC1) · Anonymous Referee #1 · 13 Apr 2016

General Comments:

I have completed my review on the manuscript "Impacts of rainfall features and antecedent soil moisture on occurrence of preferential flow: A study at hillslopes using high frequency monitoring" by Z. Peng, H. Hu, F. Tian, Q. Tie, and S. Zhao. The paper tries to answer the question: how do rainfall features and antecedent soil moisture affect the occurrence of preferential flow on different hillslopes? Generally, the paper uses a quite new technique to evaluate the occurrence of preferential flow. Although the overall results of the paper are interesting, the presentation of the paper (English,

structure of written text) is currently still lacking.

Title: second part of the title "A study at hillslopes ..." makes the overall title long and does not provide that much additional information about the content of the paper. My suggestion is to stick to a shorter version of the title.

Abstract: Some sentences are very long and make it hard to get the main message/ ideas of the paper. The paper would improve a lot if the text in the abstract is improved (some specific comments and technical comments regarding this section are given below). Additionally, the abstract should strive to more clearly summarize what the impact of the rainfall features + antecedent moisture conditions are on preferential flow – which factors affect the occurrence and how they affect the frequency/ occurrence of preferential flow.

Introduction: To create a stronger paper that more clearly demonstrates its "innovation" in the field, I would strongly suggest the authors to more rigidly convey the current limitations of previous research and the added role this paper plays to the discussion. Additionally it would help to clearly state what hypothesis you have – what you expected as an outcome of your study - and how your findings aid the field. At the moment it is not clearly stated what new concepts/ideas etc. are used (although the method used is e.g. not yet a standard method).

Introduction (2): Related to a comparison of results found in the literature: it makes it easier for the reader to generally know the methods used to analyze preferential flow in all examples. Sometimes the authors do this (e.g. mentioned that a column experiment was used), but critical information about the measurement setup fails (sensors used, dye tracers used, other tracers used?). Allaire et al (2009) - Quantifying preferential flow in soils: A review of different techniques – wrote a whole review on all techniques that could be used compare the results found, which can be used as a reference.

English language: currently, there are still a lot of grammar errors and strangely formulated sentences in the manuscript that make the manuscript less easy to read. Authors

are advised to ask help from one or more native speakers to improve the level of English of the overall manuscript. - Authors differ between a rate based and a sequence based classification of preferential flow. I think it is important to realize that the extra use of a wetting velocity provides only an additional method to detect preferential flow. The method itself is not suitable to identify differences in preferential flow processes.

The separation between the results and the discussion is not clear. It seems like the results section still partly continues in the discussion part of the paper. Additionally, data is partly already discussed in the results section. When writing a separate discussion, this should only focus on the discussion of the results, not on the disruption of the results. A solution would be to (1) write a combined results and discussion section or (2) to better separate both sections and the aims of both separate sections of the paper.

Specific Comments:

Introduction:

Page 1, line 14: Please specify which frequency. I assume you relate to the frequency of preferential flow occurrence. Be more specific, otherwise this is unclear to the reader.

Page 1, line 15-16: Again, please specify that you refer to preferential flow frequency.

Page 1, line 16-17: "Antecedent soil moisture was also significantly correlated with the frequency. However, this should largely be attributed to the differences of soil moisture among sites, since varying range of soil moisture at a specific site was not wide enough to influence the frequency significantly". This is very unclear as the authors talk both about spatial (site-to site) and temporal (site specific range) soil moisture variability. I do not see how the spatial and temporal occurrence link. . .

Page 1, line 26 – 28: I do not see how preferential flow can be densely distributed in soils. It is rather a process that is occurring, which either occurs or does not at a certain moment in time.

Page 2, line 4: The authors write: "Among the many, rainfall features and antecedent soil moisture are two essential control factors ..." I noticed these ideas are used throughout the papers, forming the general framework of this paper. I think this is nice and agree with this. Nonetheless, I think it is important to define and accurately separate spatial and temporal components. Preferential flow can occur at a specific location, related to local soil moisture (and even rainfall- in case of vegetation - interception) conditions. As an example we might look at specific locations where preferential flow occurred more frequently and relate this to local conditions. At the same time, we can also look in time and specify temporal differences in precipitation and antecedent moisture, which might be related to seasonal/ climatic conditions at a specific point in time. In this case, we might look at the amount of sensor locations that responded under certain rainfall conditions.

Page 2, line 6: "Heppel divided ... intensity driven and duration driven". I think it only makes sense to make such a statement when explaining afterwards how and why he did this. Otherwise referring to this paper does not really convey a clear message and rather raises questions.

Page 2, line 13: "Wu et al. indicated ... growth rates along with increasing rainfall intensity". I think it is important to state here how this was measured, as one problem related to the use of soil moisture sensors is that changes in water content are not observed while the soil is saturated (see Graham and Lin, 2011 - Controls and frequency of preferential flow occurrence: A 175-event analysis; Wiekenkamp et al. 2016 - Spatial and Temporal Occurrence of Preferential Flow in a Forested Headwater Catchment; I even believe it is also mentioned in the Hardie (2013) paper). Nonetheless, preferential flow can still occur in reality. This is something to generally note/ keep in mind as a limitation of the method.

Methods:

Page 3, line 14; Authors obtained information about summer canopy coverage using

SPOT, August 2013). This is not sufficient to reconstruct how this information is obtained. Which satellite was used? SPOT 6? Additionally, it would be worthwhile to understand where the 98% comes from – is this the average over all pixels with in the catchment (also: specify resolution).

Page 3, line 18: "...from high to low". Please be more precise. Does this specify the height? If so, please specify that you are talking about altitude (one might confuse it with slope angles). Additionally, I wonder how the effect of canopy can be separated from the geomorphological location on the hillslope, as it seems that all non-vegetated monitoring locations are located on a relatively flat surface.

Page 3, line 21: the authors state that sensors were installed with different maximum depths .e.g. a different number of sensors per location. It would be worthwhile to know why? Was this related to the absolute depth of the soil/ the stone content in a given depth? Additionally, it would be important to know if the number of sensors influences the frequency of preferential flow as detected by the sensor response sequence.

Page 4, line 3: Authors state that soils are not that think, but afterwards mention that the soils are 0-2 meters deep. I could imagine that a 2 meter deep soil is not that shallow. To better understand if soils are generally shallow/deep, it would be important to state how deep soils are on average or what the characteristic thickness of soils is (could even be specified for different landscape positions).

Is there any information about the soil types that were found at the logger locations (using FAO or American Soil classification)? If available, it would be important to state such information here (and in the related Tables)

"Eighty four groups of soil samples..." Is this the number of soil samples or is this the number of groups – how many samples per group are there? I guess individual samples were meant here.

Page 4, line 8: Authors mention the usage of Rosetta to estimate Ks values. It would be

important to state the function(s) used in the hierarchical artificial neural network model of ROSETTA – how are the measured soil physical parameters used to calculate Ks?

Ranges in soil properties per site are referred to in the methods section and given in table 2. What do the authors think about the factors that are the most influential for preferential flow occurrence? Is the rather range of hydraulic properties, the hydraulic properties of the most upper layer, or differences in hydraulic properties within the soil profile important for preferential flow occurrence?

Page 4, line 15 – 22: The determination of a rainfall event is commonly only defined by precipitation characteristics itself. In this case, the change in soil moisture at all depths is used. Why?

Page 4, line 15 – 22 (2): The determination of a rainfall event relies on hourly thresholds. Is the original 10 minute resolution soil moisture and precipitation used for this approach or is the data aggregated to hourly values to determine the event start and end?

Page 4, line 24: Authors refer to the "Hardie et al. (2013) method". It is however unclear what type of method (the classification, mentioned later in the section or the wetting front velocities?)– What specific part of the analysis is referred to? Plus, it would be important to specify this here for reader that has not read the Hardie paper.

Page 5, line 2: Please replace "penetration velocity of the wetting front" by "wetting front velocity". Do this consequently – also for other parts in the manuscript. Additionally, one could question whether Eq. 1 needs to be written out here.

Results:

- Page 5, line 24: "Differences . . . 46.8 mm". In which time frame? An hour/event/10 minute measurement/cumulative? - Page 6, line 3: "In order to compare . . . selected". I wonder how the similarity of the events was examined. Should start and end date any of the events be the same for all rain gage locations? If not, how were "rainfall events

observed by all rain gages" selected? - Figure 2: please specify the formula used for the curve that was fitted. And what was the RMSE of this fitted curve? In the text, the authors mention that they used a Pearson III curve. Please specify what type of curve is meant (I do not consider this a standard method). - Page 6, line 12: Differences between rainfall features were tested against the Gaussian distribution. Why? - Figure 4: I would prefer to see the data values in a table as it is difficult to infer the exact significance between sites. A table will additionally provide extra information (exact values). - 'Considering the rainfall events: Overall, it is not well specified which rainfall characteristics are used for a specific event. Where the average characteristics for all location used or are the location specific rainfall characteristics considered? Additionally: which rainfall data was used for the monitoring sites where no rainfall was measured? - Regarding figure 6: During both monitoring periods, the FH locations had several situations in which they all reacted preferentially. However, there is no situation in which all 12 sensor locations reacted at the same time. This would be an interesting point to bring up and discuss. Additionally, it is not that clear that the top four bars belong to period 1 and the lower 12 to period 2. To improve this, such information could be directly added in the figure. - On the statistics similarity of rainfall: only the similarity of the rainfall characteristics during the 39 simultaneous events was tested. Nonetheless, the events that were not occurring at the same time amongst all sites and that created local differences were not considered. Although these additional events/ variation is number of events do inform us about rainfall heterogeneities, they were "kicked out". Is it fair to afterwards state that precipitation differences did not influence the occurrence of preferential flow, although they might generate local differences amongst locations e.g. antecedent soil moisture conditions, canopy wetness.

Discussion:

Figure 7: It is not clear if the rainfall features vs. type of flow included all the specific rainfall amount for all individual events * the individual sites. If this is the case, it is logical that there is an overlap in characteristics, as Figure 6 already shows that individual

locations during the same event might cause different responses, which explains why similar rainfall conditions end up in the different classes.

Page 8, line 5:" The values ranges . . .calculated". Frequencies here probably refer to the total number of sites that responded preferentially. It is important to mention such information specifically – e.g. if you integrated the data over time (to look spatially) or in space (to look temporally). Examples of papers that apply such approaches can be found in Liu and Lin (2015 - Frequency and control of subsurface preferential flow occurrence in the Shale Hills catchment: from Pedon to catchment scales).

Page 8, line 22: "..., n =233". Where does this n value come from? Where location specific rainfall conditions connected? Again, it is not clear how rainfall conditions were used – site specific or only event specific? The way this data is used should be better described throughout the manuscript.

Page 8, section "On the influence of antecedent soil moisture". Again, here it is important to state when antecedent soil moisture conditions were spatially or temporally used. . .

Figure 11, page 20: Why is the antecedent moisture not directly plotted against the frequency of preferential flow? This would better show the relationship between both variables.

Page 16, line 5 and page 20, line5: In these different figures (5 and 12), you visualize the soil moisture response to rainfall at different depths. Be consequent and use the same color scheme for both images. Generally, I think it is more difficult to follow the legend in the black and white images (what is what). Therefore, I would suggest either using different grey-tones or sticking to the colored figures.

Technical Corrections:

Introduction: technical comments were detailed described for the introduction (as a start and an example). Authors should however ask aid from a native speaker to check
the manuscript more detailed.

Page 1, line 9: rephrase "observation was conducted . . .." Additionally, you might need to specify what observations; this is not clear in this sentence, and it is part of the abstract, sentences need to be very clear (this is the part that is most read).

Page 1, line 12 "5-10 times of the saturated . . ." Remove "of".

Page 1, line 9(end) rephrase "Totally . . ."

Page 1, line 13: change "in average" to "on average".

Page 1, line 19 – 22: "Further examination suggested that topography and surface cover . . . preferential flow". This sentence is too long and there are unclear connections – why does the sentence end with soil moisture where it started with the factors surface cover and topography?

Page 2, line 18: "though Hardie et al. (2013) suggested . . ." This sentence build-up creates the idea that although Hardie et al. (2013) suggested it, other researchers do not agree. However, the agreement between authors/researchers is not in any way related to what Hardie et al. (2013) suggested. Please rewrite.

Methods/Results:

The authors use the word "rainfalls" multiple times in the manuscript (page 15, caption Figure 2, 3.1 results). There is no such thing as "rainfalls", as the plural form or "rainfall" does not exist. Please rephrase this throughout the manuscript.

---

## Referee Comment (RC2) · Anonymous Referee #2 · 4 May 2016

Major comments: 1. The knowledge gap on preferential flow paths that necessitates the current study is not clear. 2. The introduction appears to be a summary of previous studies on preferential flow occurrence. A persuasive introduction with a clear hypothesis or research question to improve upon the previous studies should be included in Introduction. 3. From the results, it appears that frequency of the occurrence of preferential flow was not correlated to antecedent soil moisture content variation (very small) within a site, but it has a correlation when the frequency is compared between different sites. This result suggests that a small variation in antecedent moisture content within a site can trigger preferential flow. Is there a thresh-hold moisture content that

triggers preferential flow in a site? Between sites, there could be different confounding factors, so I wonder how useful it is to make a general statement about the relationship between antecedent moisture content and frequency of preferential flow. Minor comments 4. Page 1, line 9: "Observation was conducted" is vague and it sounds confusing. Be specific. 5. Page 3, line 8: What observations were carried out? Be specific. Or write a non-vague sentence. E.g., we monitored rainfall intensity, runoff, soil moisture content... at the catchment. 6. Page 4, line 6: Titration method - need a reference that describes more about the method. 7. Page 4, line 11: Data showed. Where to find those data? Be more specific about which data the readers should look for here. 8. Page 8, line 1: Grammar issue.
* * *

---

## Referee Comment (RC3) · Anonymous Referee #3 · 24 May 2016

General comments:

Identifying the factors and mechanisms leading to preferential flow of water, solutes and suspended particles through the soil is a challenging research topic, and a matter of considerable significance as it can impact the quantity and quality of the rainfall or irrigation water reaching the groundwater. Many studies attempted to identify the impact of soil physical (e.g. macropore) and chemical (e.g. water repellency) heterogeneities on the onset of preferential flow. The paper under review aims at identifying – at the hillslope scale - the impact of boundary conditions linked to rain characteristics

(called 'rainfall features' in the paper) as well as one initial condition (antecedent soil moisture) on the onset of preferential flow. This is a topic of great interest for the scientific community interested in mass transfer in soils, and it falls well within the scope of HESS.

The paper major issue is that it is difficult to understand its novelty compared to already published studies. To which extent does it go farther than previous work on high frequency monitoring of preferential flow? One reason is that the introduction is poorly written. It does survey some literature results on hillslope scale monitoring of the occurrence of preferential flow, but fails to pinpoint the gaps and opened questions. This leads to a lack of precise scientific question to address in the paper. Was this work only a mere case study? This may be fine, but, if so, this should be clearly stated. A second reason is that, although the manuscript contains a discussion section, the experimental findings are not thoroughly discussed and compared to previous finding and scientific gaps. The current discussion section is a mere continuation of the result section.

In addition, the paper is difficult to read and understand because sentences are often awkward (e.g. page 9 lines 13-16), the wording imprecise, or the language register inappropriate for a scientific paper (e.g. 'bunch' is a rather informal noun). I advise the authors to seek the help of a native English speaker to address this issue.

Still, the amount of data collected in this case study is impressive and valuable for the community. It may be useful for future use to present in the supporting information section the hydraulic conductivity for each soil layer, as a function of depth, as well as the velocity of the wetting front for each rainfalls.

Specific comments:

1/The paper relies on two criteria to determine the occurrence or absence of preferential flow, based on (i) the non-sequential response of probes with depths and (ii) the velocity of the wetting front compared to some arbitrary threshold, 5 or 10 times the

hydraulic conductivity, depending on the depth. Although similar criteria have already been used in another paper (Hardie et al. 2013), they are not backed by any theoretical developments and their capacity to discriminate between preferential and equilibrium flow is not established. Non-sequential response of probes may arise from lateral infiltration of water, especially when the soil surface is not horizontal. In addition, (1) the wetting front velocity thresholds are quite arbitrary, and (2) since the threshold varied with depth, it is not clear from lines 7-15 page 5 when preferential flow was assumed to occur: was it when the wetting front velocity was higher than the thresholds at all the depths investigated? or at only one depth ? Other criteria have been proposed to establish the occurrence of preferential flow, for example, when the rainfall intensity exceeds the infiltrability of the matrix, the exceeding water flux is likely to participate to surface run-off , or, if macropores are present, to be involved into macropore flow (Nimmo, Vadose Zone Journal 2016, doi:10.2136/vzj2015.05.0079).

2/Page 7, line 6-12: were the spatial variations of the preferential flow frequency correlated with the spatial variations of the saturated hydraulic conductivity ? or with the ratio rainfall intensity/saturated hydraulic conductivity ? It may be interesting along with figure 6 to present, with a similar color code, (i) vertically, for each site: the average, minimum and maximum hydraulic conductivity, and (ii) horizontally, for each rainfall event the rainfall amount, duration maximum and average intensities.

3/What were the local topography of each site (e.g. swale, convex , slope. . .) ? Is there an influence of the local topography on the occurrence of preferential flow at each site as noted by Liu and Lin 2015 (SSSAJ 79, 362) ? Burrowing animal such as earthworms have been shown to affect the occurrence of preferential flow (e.g. Capowiez et al., 2014 Pedobiologia, 57, 303). Could their local density explain variations of preferential flow occurrence from one site to others ?

4/Figure 7: it may have been interesting to use the so-called 'violin-plots' to represent these data.

5/ When discussing the relationship between the average soil moisture and the frequency of preferential flow (figure 10), the authors indicate that the behavior of the graph is dominated by the contrasting soil moisture content of Slope I sites at the one end, and FH3 andFH4 sites at the other end. This unequal distribution of the sites on the abscissa of figure 10 is indeed important information when interpreting the figure. I wonder if the sites were equally distributed on the abscissas of the graphs shown in figure 8. An easy way to add this information to figures 8 and 10 would be to use stacked column charts.

6/ What were the values of the real and imaginary parts of the refractive index used to determine the particle size distribution by light scattering?

Specific comments:

Page 1: line 8: most of the time 'in order to' can be simplified to 'to'. Page 2, lines 18-27: This sections is unclear and difficult to understand, probably because (i) the sentences are too long and (ii) the ideas developed in this paragraph are not well organized (e.g in the same sentence (starting line 20 and ending line 24, both the influence on preferential flow of initially wet and initially dry soils are discussed, but it is difficult to understand exactly which arguments refer to which situation)

Page 4 line 16-17: "rainfall events were divides into ines….. rains". I was not able to understand this sentence.

Page 8, Line 1-2 I was not able to understand this sentence.

---

## Referee Comment (RC4) · Anonymous Referee #4 · 3 Jun 2016

General Comments

The study of "Impacts of rainfall features and antecedent soil moisture on occurrence of preferential flow: A study at hillslopes using high frequency monitoring" is of considerable significance to the scientific community interested in better understanding the onset of preferential flow. The vast amount of observational data collected by the authors is impressive. However, a significant revision needs to be done before the manuscript becomes suitable for publication.

One of the main issues with the manuscript is that it fails to engage the reader as

to the importance of this study and how it differs from previous studies. The authors should highlight it in the Introduction section. Secondly, having a separate Discussion section is not the best approach here. Having a Results and Discussion section merged together will give the readers a better understanding of the similarities and disparities between the current study and the previous studies.

There are also a lot of grammatical mistakes in the manuscript, which makes the manuscript harder to understand and keep the focus on the actual research part. I suggest the authors have a native English speaking person review the manuscript to improve its overall quality.

Specific Comments

The title of paper can be shortened. Consider leaving out the second part of the title, as it does not provide much additional information.

Authors have used the word "rainfalls" at numerous instances throughout the manuscript. It should be "rainfall".

Missing citations in the Reference section. For example, Vogel et al. 2010; Niu, 2003. Please re-check and make sure all the references listed.

Page 1, Line 14: Remove "," after intensity.

Page 1, Line 22: Replace "knowledge" with "finding". Using the word "finding" instead of "knowledge" tells the reader the importance of the study.

Page 1, Line 26: Indent the paragraph.

Page 2, Line 4: Rewrite 1st sentence. "Among the many" what? Cite the studies after you have addressed what are "among the many".

Page 2, Line 6: Do not begin the sentence with "And". Reword the sentence.

Page, Line 20: A 2008 study, in my opinion is not "recent". Rewrite the sentence and

try not to refer a 2008 study as "recent" or cite another relevant "recent" study.

Page 2, Line 20-24: Try to break the sentence "On one hand . . . . . ." It is too long.

Page 2, Line 20: swap "became" with "becomes".

Page 2, Line 24: rewrite the sentence "However, since larger . . . . . .". It is poorly worded and is difficult to understand.

Page 2, Line 28: "Among the numerous approaches" has only reference. Please cite additional studies if "numerous approaches" are mentioned.

Page 2, Line 29: Use a semicolon instead of a comma after 1996 to cite two different papers.

Page 2, Line 3: In the last paragraph of the Introduction, the authors should not only mention what they are doing in the study but why and also highlight how their study is different from previous studies.

Page 2, Line 8: Indent the first paragraph.

Page 2, Line 16: Indent the first paragraph. I have noticed this error through out the manuscript. Please fix this.

Page 3, Line 20: The authors have not mentioned the reason as to why the probes were buried at different depths at different sites. Was it due to varying soil depths at these locations? What was the need to bury the probes further than 60cm if the authors only use 0-60cm for their analysis shown in the Figures?

Page 4, Line 11: Which "data" are the authors referring to?

Page 2, Line 27: Swap "was" with "is".

Page 4, Line 25: "Producer" is not an appropriate word here. May be the authors can use "manufacturer".

Page 4, Line 27: ". . . a bunch of studies". Use another way of describing like " a lot of

studies" or "other studies".

Page 5, Line 21: change ". . .. as was shown" to "as is shown".

Page 6, Line 4: Wrong usage of word "respectively".

Page 8, Line 22: What does n=233 refer to?

---

## Short Comment (SC1) · 4 Jun 2016

This paper tried to evaluate the impact of rainfall features and antecedent soil moisture on occurrence of preferential flow on slope in north China by interpreting response of soil moisture to rainfall. The result showed that occurring frequency of preferential flow was averagely 40.7Nevertheless, it is my feeling that the authors did not stress enough the limitation of previous researches and their relations with the major objectives. And I also concern about the method that the authors used to analyze the correlation between rainfall features, antecedent soil moisture and frequency of preferential flow. It

is not clear to me that how each rainfall feature was divided into 15 sub-ranges with non-uniform increments (P8L5). In my opinion, the method would determine the fitting curves and R2, and so the results depend on the inclination of the authors. Therefore, the authors should provide more explanation. As a general comment, I think that the paper requires major revision before being published.

I have listed in the following a number of issues that should be addressed in this paper before publication: 1. "Eighty four groups of soil samples were collected from the profiles at all sites 5 except FH4. "(P4L5). The reason should be explained why FH4 was excluded. 2. What is the accuracy of the rain gauges? 3. The slope gradient and aspect, canopy coverage and elevation of each site are suggested to add to Table1, which will help readers to understand the differences of the sites. And more explanations should be given why the sites S1H1-S1H5 and S2H1-S2H3 were set, which seem very close to each other according to Fig1. 4. What the measurement radius of the probes of TDR? The information is important because only the preference flow occurred in this range could be interpreted by the variation of observed soil moisture. 5. What is theoretical basis that a 0.002 cm3/cm3 threshold was set to quantify the responses of water content to infiltration according to a bunch of studies, given accuracies of the applied TDR probes were 0.025cm3/cm3. Whether did previous studies (Blume et al., 2009; Lin and Zhou, 2008) use the TDR probes of same accuracy? 6. The null hypothesis of the Kolmogorov-Smirnov test is usually defined as that the sample is drawn from the reference distribution (in the one-sample case) or that the samples are drawn from the same distribution (in the two-sample case) (such as in XLSTAT). However, "significant difference between every two distribution was set as the null hypothesis in this paper"(Line102-103). Which software was used to carry out the tests? 7. It is difficult to read Fig4 and I suggest change it to a table. 8. "Contents of organic matter, clay and silt generally decreased with depth, leading to higher saturated hydraulic conductivity. Detailed information at each depth was not listed in Table 2, but was covered by the ranges."(P4L11-12). What is the sampling depth of the data in Table 2? 9. The rainfall amount difference between site FH3 and S2H1 is larger

than 180mm from 2014/8/22 to 2014/10/31 (Table3) but the distance seems only about 500m. Is it because the logger at S2H1 failed from 2014/10/14 to 2014/10/31(P5L27)? This should be added as notes of Table3. 10. Fig1 is not contour map but a DEM map.

---

## Referee Comment (RC5) · Y. Zhang (Referee) · 10 Jun 2016

This paper tried to evaluate the impact of rainfall features and antecedent soil moisture on occurrence of preferential flow on slope in north China by interpreting response of soil moisture to rainfall. The result showed that occurring frequency of preferential flow was averagely 40.7% and the authors concluded that rainfall amount was the dominant driven factor to occurrence of preferential flow, while average intensity was in second. Although the conclusions are not innovative, the field observation data is reliable and results of this paper are valuable to further understand the mechanism of preferential

flow occurrence.

Nevertheless, it is my feeling that the authors did not stress enough the limitation of previous researches and their relations with the major objectives. And I also concern about the method that the authors used to analyze the correlation between rainfall features, antecedent soil moisture and frequency of preferential flow. It is not clear to me that how each rainfall feature was divided into 15 sub-ranges with non-uniform increments (P8L5). In my opinion, the method of dividing would determine the fitting curves and $R^2$, and so the results depend somewhat on the inclination of the authors. Therefore, the authors should provide more explanation to show the method is valid.

As a general comment, I think that the paper requires major revision before being published.

I have listed in the following a number of issues that should be addressed in this paper before publication.

1. "Eighty four groups of soil samples were collected from the profiles at all sites 5 except FH4. "(P4L5). The reason should be explained why FH4 was excluded.

2. What is the accuracy of the rain gauges?

3. The slope gradient and aspect, canopy coverage and elevation of each site are suggested to add to Table1, which will help readers to understand the differences of the sites. And more explanations should be given why the sites S1H1-S1H5 and S2H1-S2H3 were set, which seem very close to each other according to Fig1.

4. What the measurement radius of the probes of TDR? The information is important because only the preference flow occurred in this range could be interpreted by the variation of observed soil moisture.

5. What is theoretical basis that a 0.002 cm3/cm3 threshold was set to quantify the responses of water content to infiltration according to a bunch of studies, given accuracies of the applied TDR probes were 0.025cm3/cm3. Whether did previous studies

(Blume et al., 2009; Lin and Zhou, 2008) use the TDR probes of same accuracy?

6. The null hypothesis of the Kolmogorov-Smirnov test is usually defined as that the sample is drawn from the reference distribution (in the one-sample case) or that the samples are drawn from the same distribution (in the two-sample case) (such as in XLSTAT). However, "significant difference between every two distribution was set as the null hypothesis in this paper"(Line102-103). Which software was used to carry out the tests?

7. It is difficult to read Fig4 and I suggest change it to a table.

8. "Contents of organic matter, clay and silt generally decreased with depth, leading to higher saturated hydraulic conductivity. Detailed information at each depth was not listed in Table 2, but was covered by the ranges."(P4L11-12). What is the sampling depth of the data in Table 2?

9. The rainfall amount difference between site FH3 and S2H1 is larger than 180mm from 2014/8/22 to 2014/10/31 (Table3) but the distance seems only about 500m. Is it because the logger at S2H1 failed from 2014/10/14 to 2014/10/31(P5L27)? This should be added as notes of Table3.

10. Fig1 is not contour map but a DEM map.

---

## Editor Comment (EC1) · N. Romano (Editor) · 3 Jul 2016

Dear Authors, So far your manuscript received interesting comments and some criticisms. I think you should start providing some preliminary responses in order to get this discussion phase alive.

---

## Author Comment (AC1) · 27 Sep 2016

**General Comments:**

**I have completed my review on the manuscript "Impacts of rainfall features and antecedent soil moisture on occurrence of preferential flow: A study at hillslopes using high frequency monitoring" by Z. Peng, H. Hu, F. Tian, Q. Tie, and S. Zhao. The paper tries to answer the question: how do rainfall features and antecedent soil moisture affect the occurrence of preferential flow on different hillslopes? Generally, the paper uses a quite new technique to evaluate the occurrence of preferential flow. Although the overall results of the paper are interesting, the presentation of the paper (English, structure of written text) is currently still lacking.**

**Title: second part of the title "A study at hillslopes ..." makes the overall title long and does not provide that much additional information about the content of the paper. My suggestion is to stick to a shorter version of the title.**

Reply: Thanks for your suggestion. Given that this study is kind of a case study, we think it would be necessary to present some key information of the study area in the title. We would change the tile into *"impact of rainfall features and antecedent soil moisture on occurrence of preferential flow in a sub-humid catchment"*.

**Abstract: Some sentences are very long and make it hard to get the main message/ideas of the paper. The paper would improve a lot if the text in the abstract is improved (some specific comments and technical comments regarding this section are given below). Additionally, the abstract should strive to more clearly summarize what the impact of the rainfall features + antecedent moisture conditions are on preferential flow – which factors affect the occurrence and how they affect the frequency/ occurrence of preferential flow.**

Reply: Thanks a lot for your suggestions. We will write the sentences into shorter ones, and with the help a native English speaker, we will re-organize the presentation, to make it more readable and more professional. As you suggest, we will illustrate more on the influences of rainfall feature and antecedent soil moisture on preferential flow occurrence, as well as on relevant mechanisms.

**Introduction: To create a stronger paper that more clearly demonstrates its "innovation" in the field, I would strongly suggest the authors to more rigidly convey the current limitations of previous research and the added role this paper plays to the discussion.**

**Additionally it would help to clearly state what hypothesis you have – what you expected as an outcome of your study - and how your findings aid the field. At the moment it is not clearly stated what new concepts/ideas etc. are used (although the method used is e.g. not yet a standard method).**

**Introduction (2): Related to a comparison of results found in the literature: it makes it easier for the reader to generally know the methods used to analyze preferential flow in all examples. Sometimes the authors do this (e.g. mentioned that a column experiment was used), but critical information about the measurement setup fails (sensors used, dye tracers used, other tracers used?). Allaire et al (2009) - Quantifying preferential flow in soils: A review of different techniques – wrote a whole review on all techniques that could be used compare the results found, which can be used as a reference.**

Reply: Thanks a lot for your suggestions on organizing the introduction and presenting the literatures. We will illustrate more to fill the gap between previous studies and our objectives, and state more clearly about the necessity of this study.

Basically, this study was initiated from two considerations. (1) It would be helpful in understanding the processes of subsurface hydrology, if we get the key factors that control the occurrence of preferential flow. Lots of studies have been carried out on this topic. However, contradictory results were obtained in different cases, e.g., the cases of Wu et al. (2014) and Hardie et al. (2013). And to our knowledge, no study on this topic has been carried out in northern China with sub-humid climate and poorly developed underlying soil. Hence, we think this study could be a complementary to the understanding of controlling factors of preferential flow; meanwhile, it would be helpful in understanding hydrological processes of the study area. (2) By far, there are many methods for the detection of preferential flow, but in-suit method is rather limited. The method using wetting front as an indicator, which was proposed by Lin and Zhou (2008)

and later improved by Hardie et al. (2013), could be an alternative option. Since this method has been on applied in only two or three cases to our knowledge, it would be of interest to apply it in our study area, where climate and surface condition are different from previous cases.

**English language: currently, there are still a lot of grammar errors and strangely formulated sentences in the manuscript that make the manuscript less easy to read. Authors are advised to ask help from one or more native speakers to improve the level of English of the overall manuscript.**

Reply: Thanks a lot for your comments. We will solve the Language problems with help from a Native English speaker.

**Authors differ between a rate based and a sequence based classification of preferential flow. I think it is important to realize that the extra use of a wetting velocity provides only an additional method to detect preferential flow. The method itself is not suitable to identify differences in preferential flow processes. The separation between the results and the discussion is not clear. It seems like the results section still partly continues in the discussion part of the paper. Additionally, data is partly already discussed in the results section. When writing a separate discussion, this should only focus on the discussion of the results, not on the disruption of the results. A solution would be to (1) write a combined results and discussion section or (2) to better separate both sections and the aims of both separate sections of the paper.**

Reply: We agree that the rate based method is an additional method to the sequence based method, since preferential flow detected by the two methods may occurs at the same time and cannot be absolutely separated. In this study, we classify the preferential as PF-ns as long as non-sequential responses occur. In this way, we think it would be of interest to compare between the preferential flow that shows sequential responses and non-sequential responses.

Thanks for your suggestions on re-organizing the sections. We will combine Section 3 and Section 4 together as a section of "Results and Discussion". We will make more comparisons between our results with those of the previous studies, so as to make our results sounder and more meaningful.

**Specific Comments:**

**Introduction:**

**Page 1, line 14: Please specify which frequency. I assume you relate to the frequency of preferential flow occurrence. Be more specific, otherwise this is unclear to the reader.**

Reply: In fact, all the words of "frequency" in the abstract refer to "frequency of preferential flow occurrence". It will be clarified in the revised abstract.

**Page 1, line 15-16: Again, please specify that you refer to preferential flow frequency.**

Reply: Thanks for your suggestions. It will be clarified in revision.

**Page 1, line 16-17: "Antecedent soil moisture was also significantly correlated with the frequency. However, this should largely be attributed to the differences of soil moisture among sites, since varying range of soil moisture at a specific site was not wide enough to influence the frequency significantly". This is very unclear as the authors talk both about spatial (site-to site) and temporal (site specific range) soil moisture variability. I do not see how the spatial and temporal occurrence link. . .**

Reply: We are sorry for the unclear statement. We will improve the presentation in revision. Generally, points of this sentence is as follow:

If the data of all sites were taken as a whole, a significantly correlation between frequency of PF and antecedent soil moisture would be obtained (see Figure 10).

However, soil moisture was continuously lower at some sites than at other sites. Meanwhile, PF occurred more frequently at sites with higher soil moisture (See Figure 11).

There is a chance that an unknown factor, which induces occurrence of PF, is happened to be more significant at sites with higher soil moisture.

In order to exclude this possibility, we need to analysis the correlation between soil moisture and frequency of PF at a specific site. However unfortunately, the varying range of soil moisture at a specific is not wide enough, and no significant correlation could be obtained.

**Page 1, line 26 – 28: I do not see how preferential flow can be densely distributed in soils. It is rather a process that is occurring, which either occurs or does not at a certain moment in time.**

Reply: Thanks for your comment. We were meant to state that "macropores are densely distributed…" initially. We will delete the inappropriate statement in revision.

**Page 2, line 4: The authors write: "Among the many, rainfall features and antecedent soil moisture are two essential control factors . . ." I noticed these ideas are used throughout the papers, forming the general framework of this paper. I think this is nice and agree with this. Nonetheless, I think it is important to define and accurately separate spatial and temporal components. Preferential flow can occur at a specific location, related to local soil moisture (and even rainfall- in case of vegetation - interception) conditions. As an example we might look at specific locations where preferential flow occurred more frequently and relate this to local conditions. At the same time, we can also look in time and specify temporal differences in precipitation and antecedent moisture, which might be related to seasonal/ climatic conditions at a specific point in time. In this case, we might look at the amount of sensor locations that responded under certain rainfall conditions.**

Reply: Thanks a lot for your detailed suggestion. It would be a notable improvement to categorize the factors into spatial and temporal components. However, we are afraid that it may not be robust enough to classify meteorology as temporal components, or to classify soil moisture as time-invariant spatial component. Especially in our case, rainfall is spatially heterogeneous, and soil moisture temporally variable, though not very significantly.

**Page 2, line 6: "Heppel divided . . . intensity driven and duration driven". I think it only makes sense to make such a statement when explaining afterwards how and why he did this. Otherwise referring to this paper does not really convey a clear message and rather raises questions.**

Reply: Thanks a lot for your suggestion. We will illustrate more about Heppel's study, and according to your suggestion above as well, we will re-organize the literature review work of the introduction.

**Page 2, line 13: "Wu et al. indicated . . . growth rates along with increasing rainfall intensity". I think it is important to state here how this was measured, as one problem related to the use of soil moisture sensors is that changes in water content are not observed while the soil is saturated (see Graham and Lin, 2011 - Controls and frequency of preferential flow occurrence: A 175-event analysis; Wiekenkamp et al. 2016 – Spatial and Temporal Occurrence of Preferential Flow in a Forested Headwater Catchment; I even believe it is also mentioned in the Hardie (2013) paper). Nonetheless, preferential flow can still occur in reality. This is something to generally note/ keep in mind as a limitation of the method.**

Reply: Thanks for your reminding. We will present previous studies in a more detailed and more logical way.

**Methods:**

**Page 3, line 14; Authors obtained information about summer canopy coverage using SPOT, August 2013). This is not sufficient to reconstruct how this information is obtained. Which satellite was used? SPOT 6? Additionally, it would be worthwhile to understand where the 98% comes from – is this the average over all pixels with in the catchment (also: specify resolution).**

Reply: We are sorry that we misinformed the source of the canopy coverage information. It should be corrected that the original source is the image from satellite WORLDVIEW-2, and resolution of the image is 1.0m. The image was processed by the Twenty First Century Aerospace Technology Co.,Ltd. (http://www.21at.com.cn/en/), who provided us with the information about canopy coverage.

It should be clarified that 98% is an average coverage of canopy all over the catchment, since some areas in the catchment is free of canopy cover all over the year.

We will add the information above during revision.

**Page 3, line 18: ". . .from high to low". Please be more precise. Does this specify the height? If so, please specify that you are talking about altitude (one might confuse it with slope angles). Additionally, I wonder how the effect of canopy can be separated from the geomorphological location on the hillslope, as it seems that all non-vegetated monitoring locations are located on a relatively flat surface.**

Reply: "…from high to low" means altitude here. Altitude of each site will be added into Table 1 in revision and the statement will be specified.

Concerning on the canopy coverage, sites S1H1- S1H5 and sites S2H1-S2H3 are located at two typical hillslopes, and both slopes are well covered by canopy, while sites FH1-FH4 are close to 4 meteorological stations, respectively, thus these locations are relatively flat and free of canopy cover.

**Page 3, line 21: the authors state that sensors were installed with different maximum depths .e.g. a different number of sensors per location. It would be worthwhile to know why? Was this related to the absolute depth of the soil/ the stone content in a given depth? Additionally, it would be important to know if the number of sensors influences the frequency of preferential flow as detected by the sensor response sequence.**

Reply: We are sorry the unclear statement. The probes are installed at different depths for other purposes, while in this study, only the data of the 0-60cm was used, so as to compare between the sites. We will make additional statements to clarify this issue.

**Page 4, line 3: Authors state that soils are not that think, but afterwards mention that the soils are 0-2 meters deep. I could imagine that a 2 meter deep soil is not that shallow. To better understand if soils are generally shallow/deep, it would be important to state how deep soils are on average or what the characteristic thickness of soils is (could even be specified for different landscape positions). Is there any information about the soil types that were found at the logger locations (using FAO or American Soil classification)? If available, it would be important to state such information here (and in the related Tables) "Eighty four groups of soil samples. . ." Is this the number of soil samples or is this the number of groups – how many samples per group are there? I guess individual samples were meant here.**

Reply: Soils are poorly developed in the study area. They are mostly mixed with gravels and bedrock is occasionally exposed at the surface. To our experiences of soil sampling, it was usually hard to drill deeper than 60cm by maual tools, but we digged deeper than 100cm at some locations. So we think soil depth is highly spatially heterogeneous, but generally shallow.

We will extend Table 2 to include soil properties of each layer. Soil types will be classified by the FAO standard, and will be added in to Table 2.

It should be clarified that the "84" refers to the number of group. A group means soil samples collected at a depth of site, and 3 samples were collected in each group.

**Page 4, line 8: Authors mention the usage of Rosetta to estimate Ks values. It would be important to state the function(s) used in the hierarchical artificial neural network model of ROSETTA – how are the measured soil physical parameters used to calculate Ks? Ranges in soil properties per site are referred to in the methods section and given in table 2. What do the authors think about the factors that are the most influential for preferential flow occurrence? Is the rather range of hydraulic properties, the hydraulic properties of the most upper layer, or differences in hydraulic properties within the soil profile important for preferential flow occurrence?**

Reply: To our knowledge, the ROSETTA software has been trained by a big dataset of soil properties, and its predictions are reasonably reliable. The soil properties we used to calculate Ks include bulk density, percentage of clay, silt and sand particles.

Unfortunately, we did not think about the influence of soil properties on preferential flow occurrence. But thanks to your questions, we agree that it's of importance in studying preferential flow. We will try to have a discussion on this issue base on our data, and will present the result if it is sound.

**Page 4, line 15 – 22: The determination of a rainfall event is commonly only defined by precipitation characteristics itself. In this case, the change in soil moisture at all depths is used. Why?**

Reply: This is a special requirement of this study. Since preferential flow is indicated by the responses of probes, we will not be able to confirm the responses are caused by rainfall if measurements of the probes are continuously changing before the rain.

**Page 4, line 15 – 22 (2): The determination of a rainfall event relies on hourly thresholds. Is the original 10 minute resolution soil moisture and precipitation used for this approach or is the data aggregated to hourly values to determine the event start and end?**

Reply: The original data are used to determine start and end. Since methods used to detect preferential flow are based on high frequency monitoring, we cannot afford to lose any temporal resolution of the data.

**Page 4, line 24: Authors refer to the "Hardie et al. (2013) method". It is however unclear what type of method (the classification, mentioned later in the section or the wetting front velocities?)– What specific part of the analysis is referred to? Plus, it would be important to specify this here for reader that has not read the Hardie paper.**

Reply: Thanks for your suggestion. Both the classification of preferential flow (i.e., PF-ns and PF-rate) and the methods to detect preferential flow are referred to Hardie et al. (2013). We will specify the statement in revision.

**Page 5, line 2: Please replace "penetration velocity of the wetting front" by "wetting front velocity". Do this consequently – also for other parts in the manuscript. Additionally, one could question whether Eq. 1 needs to be written out here.**

Reply: Thanks a lot. We will use the phrase "wetting front velocity" in the manuscript. Concerning on Eq.(1), we think it's concise way to illustrate how the wetting front velocity was obtained in this study, though equation is quite simple.

**Results:**

**Page 5, line 24: "Differences . . . 46.8 mm". In which time frame? An hour/event/10 minute measurement/cumulative?**

Reply: The number "46.8 mm" is a cumulative difference based on 10 minute measurement at the two sites. We will clarify this issue in revision.

**Page 6, line 3: "In order to compare . . . selected". I wonder how the similarity of the events was examined. Should start and end date any of the events be the same for all rain gage locations? If not, how were "rainfall events observed by all rain gages" selected?**

Reply: Yes, the start and end time of the selected events are the same for the rain gauges. We will improve the presentation to make it clearer.

**Figure 2: please specify the formula used for the curve that was fitted. And what was the RMSE of this fitted curve? In the text, the authors mention that they used a Pearson III curve. Please specify what type of curve is meant (I do not consider this a standard method). –**

Reply: We are sorry that we may have miswritten the name of curve. It should be more properly named as Pearson type III distribution. It is a gamma distribution, and the parameters can be obtained from the skewness (Cs) and the coefficient of variation (Cv) of the data series. We will illustrate more about the fitting curve in revision.

**Page 6, line 12: Differences between rainfall features were tested against the Gaussian distribution. Why?**

Reply: In the beginning, we aimed to confirm that the distribution of rainfall features were significantly different from the Gaussian distribution. But in the second view of the manuscript, this argument appears to be not that necessary in interpretating data. We will delete this comparison in revision.

**Figure 4: I would prefer to see the data values in a table as it is difficult to infer the exact significance between sites. A table will additionally provide extra information (exact values). 'Considering the rainfall events: Overall, it is not well specified which rainfall characteristics are used for a specific event. Where the average characteristics for all location used or are the location specific rainfall**

**characteristics considered? Additionally: which rainfall data was used for the monitoring sites where no rainfall was measured?**

Reply: Thanks for your suggestion. We will change Figure 4 to a Table, to have all the values been clearly presented.

In this study, when a monitoring site is no installed with a rain gauge, measurements of the nearest rain gauge were used for this site.

**Regarding figure 6: During both monitoring periods, the FH locations had several situations in which they all reacted preferentially. However, there is no situation in which all 12 sensor locations reacted at the same time. This would be an interesting point to bring up and discuss. Additionally, it is not that clear that the top four bars belong to period 1 and the lower 12 to period 2. To improve this, such information could be directly added in the figure. - On the statistics similarity of rainfall: only the similarity of the rainfall characteristics during the 39 simultaneous events was tested. Nonetheless, the events that were not occurring at the same time amongst all sites and that created local differences were not considered. Although these additional events/ variation is number of events do inform us about rainfall heterogeneities, they were "kicked out". Is it fair to afterwards state that precipitation differences did not influence the occurrence of preferential flow, although they might generate local differences amongst locations e.g. antecedent soil moisture conditions, canopy wetness.**

Reply: Thanks for your suggestion. We will add temporal information to the two parts of Figure 6.

If we understand correctly, the reviewers means that the frequency of preferential flow is a biased result since we kicked out some rainfall event, unless the rainfall features do not influence the occurrence of preferential flow. We agree on this point. Since the kicked out rainfall events are dominantly light rains, which are less likely to induce preferential flow as presented by the results, the frequency of preferential flow should have been overestimated in our study. However, if we want to compare between different sites, we would hope to check the responses of the sites to the same or at least similar rainfall processes. This is reason why we compared the distribution of rainfall features measured at different locations (see Figure 4), and this is the way to make the results of showed by Figure 11 meaningful.

**Discussion:**

**Figure 7: It is not clear if the rainfall features vs. type of flow included all the specific rainfall amount for all individual events * the individual sites. If this is the case, it is logical that there is an overlap in characteristics, as Figure 6 already shows that individual locations during the same event might cause different responses, which explains why similar rainfall conditions end up in the different classes.**

Reply: Figure 7 shows the collective results of the responses of all sites to rainfall. Indeed, Figure 6 shows that responses of different sites to the same rainfall event could be different. But these differences should be ascribed to some other factors, such as antecedent soil moisture, slope gradient or surface cover as we discussed in the later part. Therefore, the relationship between rainfall features and frequency of preferential flow at a specific site may not be the same with that showed in Figure 7. But still, since Figure 7 shows the responses of all sites to the similar rainfall events, the results should be statistically valid.

**Page 8, line 5:" The values ranges . . .calculated". Frequencies here probably refer to the total number of sites that responded preferentially. It is important to mention such information specifically – e.g. if you integrated the data over time (to look spatially) or in space (to look temporally). Examples of papers that apply such approaches can be found in Liu and Lin (2015 - Frequency and control of subsurface preferential flow occurrence in the Shale Hills catchment: from Pedon to catchment scales).**

Reply: Thanks a lot for your suggestion. We will improve the presentation in revision, to make the illustration clearer.

**Page 8, line 22: "..., n =233". Where does this n value come from? Where location specific rainfall conditions connected? Again, it is not clear how rainfall conditions were used – site specific or only event specific? The way this data is used should be better described throughout the manuscript.**

Reply: "n" is the cumulative number of the events that observed by the 9 rain gauges and used for analysis. We will illustrate more clearly about this correlation analysis.

**Page 8, section "On the influence of antecedent soil moisture". Again, here it is important to state when antecedent soil moisture conditions were spatially or temporally used. . .**

Reply: Thanks a lot for your suggestion. We will improve the presentation in revision, to make the illustration clearer.

**Figure 11, page 20: Why is the antecedent moisture not directly plotted against the frequency of preferential flow? This would better show the relationship between both variables.**

Reply: We thought to list the names of the sites in the same order with those of other figures (e.g., Figure 6), which may help one's reading. Thanks for your suggestion, and we will plot the antecedent soil moisture versus frequency of preferential flow, and will have the names of sites be labeled close to relevant value dot.

**Page 16, line 5 and page 20, line5: In these different figures (5 and 12), you visualize the soil moisture response to rainfall at different depths. Be consequent and use the same color scheme for both images. Generally, I think it is more difficult to follow the legend in the black and white images (what is what). Therefore, I would suggest either using different grey-tones or sticking to the colored figures.**

Reply: Thanks a lot for your suggestion. We will redraw Figure 5 into a colored image, with the same color scheme with Figure 12.

**Technical Corrections:**

**Introduction: technical comments were detailed described for the introduction (as a start and an example). Authors should however ask aid from a native speaker to check the manuscript more detailed.**

**Page 1, line 9: rephrase "observation was conducted . . .." Additionally, you might need to specify what observations; this is not clear in this sentence, and it is part of the abstract, sentences need to be very clear (this is the part that is most read).**

Reply: Thanks for your suggestion. Besides the background information of the study area, e.g., topography, soil texture, canopy coverage, soil moisture was observed at 12 sites, 9 of which were equipped with rain gauges.

We will specify the "observation" in revision.

**Page 1, line 12 "5-10 times of the saturated . . ." Remove "of".**

Reply: "of" will be dropped in revision.

**Page 1, line 9(end) rephrase "Totally . . ."**

Reply: This sentence will be rephrased.

**Page 1, line 13: change "in average" to "on average".**

Reply: We will change "in average" to "on average".

**Page 1, line 19 – 22: "Further examination suggested that topography and surface cover . . . preferential flow". This sentence is too long and there are unclear connections – why does the sentence end with soil moisture where it started with the factors surface cover and topography?**

Reply: We were about the emphasis that topography and surface covers plays a more essential role than soil moisture in controlling occurrence of preferential flow. We will rewrite this sentence into short one and state more clearly.

**Page 2, line 18: "though Hardie et al. (2013) suggested . . ." This sentence build-up creates the idea that although Hardie et al. (2013) suggested it, other researchers do not agree. However, the agreement between authors/researchers is not in any way related to what Hardie et al. (2013) suggested. Please rewrite.**

Reply: Thanks for your reminding. Since Hardie's et al. (2013) statement is not necessary in this sentence, we will drop the citation of the Hardie's et al. (2013) study here.

**Methods/Results:**

**The authors use the word "rainfalls" multiple times in the manuscript (page 15, caption Figure 2, 3.1 results). There is no such thing as "rainfalls", as the plural form or "rainfall" does not exist. Please rephrase this throughout the manuscript.**

Reply: Thanks for reminding. We will check throughout the manuscript, to make sure that the word is properly used.

---

## Author Comment (AC2) · 27 Sep 2016

**Major comments:**

**1. The knowledge gap on preferential flow paths that necessitates the current study is not clear.**

Reply: Thanks a lot for your comments.

The first paragraph of introduction talks about the basic knowledge of preferential flow, its unique features compared with uniform flow, and its potential influences on hydrological processes. In this way, we sought to emphasis that preferential flow is of great importance in hydrological studies, and thus needs to be better understood. The first paragraph could be a preface of the following statements, since study on the controlling factors of preferential flow is one of the major ways to understand preferential flow.

We will improve our presentations in revision, so as to have the paragraphs more logically connected.

**2. The introduction appears to be a summary of previous studies on preferential flow occurrence. A persuasive introduction with a clear hypothesis or research question to improve upon the previous studies should be included in Introduction.**

Reply: We will illustrate more to fill the gap between previous studies and our objectives.

Basically, this study was initiated from two considerations. (1) It would be helpful in understanding the processes of subsurface hydrology, if we get the key factors that control the occurrence of preferential flow. Lots of studies have been carried out on this topic. However, contradictory results were obtained in different cases, e.g., the cases of Wu et al. (2014) and Hardie et al. (2013). And to our knowledge, no study on this topic has been carried out in northern China with sub-humid climate and poorly developed underlying soil. Hence, we think this study could be a complementary to the understanding of controlling factors of preferential flow; meanwhile, it would be helpful in understanding hydrological processes of the study area. (2) By far, there are many methods for the detection of preferential flow, but in-suit method is rather limited. The method using wetting front as an indicator, which was proposed by Lin and Zhou (2008) and later improved by Hardie et al. (2013), could be an alternative option. Since this method has been on applied in only two or three cases to our knowledge, it would be of interest to apply it in our study area, where climate and surface condition are different from previous cases.

**3. From the results, it appears that frequency of the occurrence of preferential flow was not correlated to antecedent soil moisture content variation (very small) within a site, but it has a correlation when the frequency is compared between different sites. This result suggests that a small variation in antecedent moisture content within a site can trigger preferential flow. Is there a threshhold moisture content that triggers preferential flow in a site? Between sites, there could be different confounding factors, so I wonder how useful it is to make a general statement about the relationship between antecedent moisture content and frequency of preferential flow.**

Reply: We are afraid there may be some misunderstandings here.

We don't think it can be concluded that a small variation in antecedent soil moisture at a site can trigger preferential flow. Given the limited varying range of soil moisture at a specific site throughout the monitoring period (see Figure 11), no significant correlation between the frequency of preferential flow and antecedent soil moisture can be obtained. This means that frequencies of preferential flow at various antecedent soil moistures are not significantly different from each other, at least, there is no trend of the frequency versus antecedent soil moisture. Therefore, we don't think there is a threshold of soil moisture that "triggers" preferential flow, at least the threshold is not within the varying ranges of observed soil moisture.

We agree that there may be some confounding factors between the sites, and this is the reason why we cannot draw a conclusion about the influence of antecedent soil moisture on occurrence of preferential flow, though Figure 10 shows a significant correlation between the two. The later part of Section 4.2 tries to clarify that the time-invariant components, such as slope gradient and surface cover, may have played a essential role in controlling soil moisture as well as occurrence of preferential flow at a specific site.

**Minor comments**

**4. Page 1, line 9: "Observation was conducted" is vague and it sounds confusing. Be specific.**

Reply: Thanks for your suggestion. Besides the background information of the study area, e.g., topography, soil texture, canopy coverage, soil moisture was observed at 12 sites, 9 of which were equipped with rain gauges.

We will specify the "observation" in revision.

**5. Page 3, line 8: What observations were carried out? Be specific. Or write a non-vague sentence. E.g., we monitored rainfall intensity, runoff, soil moisture content. . . at the catchment.**

Reply: Thanks for your comment. We will not talk about the "observation" in this section after revision. Instead, we will start the paragraph with "Figure 1 illustrates the Xitaizi experimental catchment, which is …", and the "observations" will be specified in the following sections.

**6. Page 4, line 6: Titration method – need a reference that describes more about the method.**

Reply: Thanks for your suggestion. This method follows the procedures suggested by the National Standard of P.R.China (GB9384-88). The procedures are generally as follow:

(1) Soil sample was air-dried before the test.

(2) 0.1-0.2g (accuracy of $\pm 0.0001$g) of soil sample was taken into a 150ml triangular flask, and was well mixed with 10ml 1mol/L $K_2Cr_2O_7$.

(3) The triangular flask was then placed into boiled water for 30min after being installed with an air set pipe at the top. And the triangular flask was moderately shacked every 5min during this period.

(4) The sample was cooled to room temperature. The air set pipe was rinsed before being uninstalled, and the leachate was collected by the triangular flask. Volume of the mixture in the flask was controlled to be 60-80ml.

(5) 3-5 drops of phenanthroline were added into the flask.

(6) The sample was then titrated by the 0.2 mol/L $FeSO_4$ solution, and the titration should be stopped when the color of the sample turned brownish red.

(7) The same procedures should be applied to a controlled group (0.500g of $SiO_2$).

(8) Finally, content of organic matter could be obtained by the follow equation:

$$X(\%) = \frac{(V_0 - V)C \times 3 \times 1.724 \times 100}{m}$$

Where X is the content of organic matter, %; $V_0$ is the volume of $FeSO_4$ solution consumed in testing $SiO_2$, ml; V is the volume of $FeSO_4$ solution consumed in testing soil sample, ml; C is the concentration of $FeSO_4$ solution used for titration, mol/ml; m is the weight of air-dried soil sample, g. Additionally, the digit 3 refers to the ¼ molar weight of carbon atoms, g/mol; and 1.724 is the coefficient that transfers the weight of organic carbon to the weight of organic matter.

**7. Page 4, line 11: Data showed. Where to find those data? Be more specific about which data the readers should look for here.**

Reply: "Data" here refers to Table 2. We will specify the illustration in revision.

**8. Page 8, line 1: Grammar issue.**
Reply: We will have a native English speaker to help us on the language issue. We will check through the manuscript, and improve the presentations in revision.

---

## Author Comment (AC3) · 27 Sep 2016

**General comments:**

Identifying the factors and mechanisms leading to preferential flow of water, solutes and suspended particles through the soil is a challenging research topic, and a matter of considerable significance as it can impact the quantity and quality of the rainfall or irrigation water reaching the groundwater. Many studies attempted to identify the impact of soil physical (e.g. macropore) and chemical (e.g. water repellency) heterogeneities on the onset of preferential flow. The paper under review aims at identifying – at the hillslope scale - the impact of boundary conditions linked to rain characteristics (called 'rainfall features' in the paper) as well as one initial condition (antecedent soil moisture) on the onset of preferential flow. This is a topic of great interest for the scientific community interested in mass transfer in soils, and it falls well within the scope of HESS.

The paper major issue is that it is difficult to understand its novelty compared to already published studies. To which extent does it go farther than previous work on high frequency monitoring of preferential flow? One reason is that the introduction is poorly written. It does survey some literature results on hillslope scale monitoring of the occurrence of preferential flow, but fails to pinpoint the gaps and opened questions. This leads to a lack of precise scientific question to address in the paper. Was this work only a mere case study? This may be fine, but, if so, this should be clearly stated.

A second reason is that, although the manuscript contains a discussion section, the experimental findings are not thoroughly discussed and compared to previous finding and scientific gaps. The current discussion section is a mere continuation of the result section.

In addition, the paper is difficult to read and understand because sentences are often awkward (e.g. page 9 lines 13-16), the wording imprecise, or the language register inappropriate for a scientific paper (e.g. 'bunch' is a rather informal noun). I advise the authors to seek the help of a native English speaker to address this issue.

Still, the amount of data collected in this case study is impressive and valuable for the community. It may be useful for future use to present in the supporting information section the hydraulic conductivity for each soil layer, as a function of depth, as well as the velocity of the wetting front for each rainfalls.

Reply: Thanks a lot for your comments.

We will illustrate more to fill the gap between previous studies and our objectives. Basically, this study was initiated from two considerations. (1) It would be helpful in understanding the processes of subsurface hydrology, if we get the key factors that control the occurrence of preferential flow. Lots of studies have been carried out on this topic. However, contradictory results were obtained in different cases, e.g., the cases of Wu et al. (2014) and Hardie et al. (2013). And to our knowledge, no study on this topic has been carried out in northern China with sub-humid climate and poorly developed underlying soil. Hence, we think this study could be a complementary to the understanding of controlling factors of preferential flow; meanwhile, it would be helpful in understanding hydrological processes of the study area. (2) By far, there are many methods for the detection of preferential flow, but in-suit method is rather limited. The method using wetting front as an indicator, which was proposed by Lin and Zhou (2008) and later improved by Hardie et al. (2013), could be an alternative option. Since this method has been on applied in only two or three cases to our knowledge, it would be of interest to apply it in our study area, where climate and surface condition are different from previous cases. From this point of view, we agree that this study could be regard as a case study. And in the revised introduction, we will emphasis more about the difference between the study area and those of the previous studies, so as to illustrate the necessity to conduct studies at this area.

We've been hesitating whether to put Section 4 as discussion or as a part of results at the beginning. Thanks for your suggestion. We will combine Section 3 and Section 4 together as a section of "results and discussion". We will make more comparisons between our results with those of the previous studies, so as to make our results sounder and more meaningful.

Concerning on the language issue, we will have a native English speaker for some help during revision. We will check the syntax errors through the manuscript, and improve the presentations.

Thanks a lot for your suggestions on presenting a supporting information section. We will extend Table 2 to include detailed information of physical properties of soils at each depth. As well, we will find a proper way to present the wetting front velocities at each depth in each rainfall event, and as you suggested, a supporting information section would be a good choice.

**Specific comments:**

**1. The paper relies on two criteria to determine the occurrence or absence of preferential flow, based on (i) the non-sequential response of probes with depths and (ii) the velocity of the wetting front compared to some arbitrary threshold, 5 or 10 times the hydraulic conductivity, depending on the depth. Although similar criteria have already been used in another paper (Hardie et al. 2013), they are not backed by any theoretical developments and their capacity to discriminate between preferential and equilibrium flow is not established. Non-sequential response of probes may arise from lateral infiltration of water, especially when the soil surface is not horizontal. In addition, (1) the wetting front velocity thresholds are quite arbitrary, and (2) since the threshold varied with depth, it is not clear from lines 7-15 page 5 when preferential flow was assumed to occur: was it when the wetting front velocity was higher than the thresholds at all the depths investigated? or at only one depth ? Other criteria have been proposed to establish the occurrence of preferential flow, for example, when the rainfall intensity exceeds the infiltrability of the matrix, the exceeding water flux is likely to participate to surface run-off ,or, if macropores are present, to be involved into macropore flow (Nimmo, Vadose Zone Journal 2016, doi:10.2136/vzj2015.05.0079).**

Reply: Regarding criteria (i), we agree that lateral infiltration may influence the responses of probes. However, it should be noticed that, (1) measurements of the probes were constant within one or two hours before rainfall, so it's reasonable to assume that the responses of the probes were caused by rainfall; (2) measurement of a probe covers an area of soils, but not a point. Therefore, if it were the lateral infiltration that caused non-sequential response, the water should have come from somewhere on the surface with a distance from the surface area right above the probe. In other words, water should have moved farther and faster in the lateral infiltration than in the vertical infiltration, no matter preferential flow occurred or not in the vertical infiltration. Therefore, it should be confident that the lateral infiltration flows through a preferred path.

Regarding criteria (ii), we think that the threshold is arbitrary to some extent, but not totally. Given the complexity of the $v_{wf}/k_s$ ratio of equilibrium flow in various conditions, we are afraid it's beyond the reach of this study to have a theoretical based threshold. In Hardie's et al. (2013) study, the threshold was rather conservative. They made sure that equilibrium flow would not readily be misjudged as preferential flow, and they got reasonable results. However, this conservative threshold may have misjudged preferential flow as equilibrium flow more often, since the wetting front velocity would decrease with depth. In view of this problem, we adjusted the threshold to lower velocity.

We are sorry that we did not state clearly about the criteria of preferential flow occurrence of a site. To be short, by our criteria, preferential flow occurs at a site as long as it occurs at one or more depths of the site.

Thanks a lot for reminding us of other criterion. However, given our limited knowledge of soil morphology in the study area, we are not able to study in-detail about the flow paths; and because of the unknown interception and storage of water by the surface cover, it may not be feasible to compare between the rainfall intensity and infiltrability of matrix in such a short time scale.

**2. Page 7, line 6-12: were the spatial variations of the preferential flow frequency correlated with the spatial variations of the saturated hydraulic conductivity? or with the ratio rainfall intensity/saturated hydraulic conductivity ? It may be interesting along with figure 6 to present, with a similar color code, (i) vertically, for each site: the average, minimum and maximum hydraulic conductivity, and (ii) horizontally, for each rainfall event the rainfall amount, duration maximum and average intensities.**

Reply: It's a wonderful suggestion, and thanks a lot. We will try on this issue in revision, and will present the result in the manuscript if it's significant.

**3. What were the local topography of each site (e.g. swale, convex , slope. . .) ? Is there an influence of the local topography on the occurrence of preferential flow at each site as noted by Liu and Lin 2015**

**(SSSAJ 79, 362) ? Burrowing animal such as earthworms have been shown to affect the occurrence of preferential flow (e.g. Capowiez et al., 2014 Pedobiologia, 57, 303). Could their local density explain variations of preferential flow occurrence from one site to others?**

Reply: We think we have some discussion on the topography issue, though not very in-detail. We compared the responses of soil moisture to the same rainfall event between site FH3 and site S1H3 (see Figure 12). The former site is located in a relatively flat area, and data shows that maximum soil water storage increment at this site exceeded the rainfall amount, while the later is located on hillslope, and only small portion of rainfall water infiltrated into soils. This could be reason why soil moisture at FH3 was continuously higher than that at S1H3, and so were the frequency of preferential flow (see Figure 11).

The spatial variation of burrowing animal density could be an explanation that could not be excluded, but it's also involved with other similar factors, e.g. the root density. Detailed inspections of these issues are needed to make quantitative analysis, but it's beyond the reach of our data currently. We agree that this issue is worth-noting, and we will try to have some discussion about it in future studies.

**4. Figure 7: it may have been interesting to use the so-called 'violin-plots' to represent these data.**

Reply: Great suggestion! We will redraw the figure using the violin plots; it will present the density distribution of the values much more clearly. Thanks a lot.

**5. When discussing the relationship between the average soil moisture and the frequency of preferential flow (figure 10), the authors indicate that the behavior of the graph is dominated by the contrasting soil moisture content of Slope I sites at the one end, and FH3 andFH4 sites at the other end. This unequal distribution of the sites on the abscissa of figure 10 is indeed important information when interpreting the figure. I wonder if the sites were equally distributed on the abscissas of the graphs shown in figure 8. An easy way to add this information to figures 8 and 10 would be to use stacked column charts.**

Reply: Thanks a lot. It will be easier to show the contribution of each site to the frequency of preferential flow.

**6. What were the values of the real and imaginary parts of the refractive index used to determine the particle size distribution by light scattering?**

Reply: The particle analysis was done by another group of people several years ago, and they did not record the settings. But as we can remember, the values were set differently for each sample to fit the data. Values of the real part were generally between 1.5~2.0, and the values of the imaginary parts were 0.01i~0.1i.

**Specific comments:**

**Page 1: line 8: most of the time 'in order to' can be simplified to 'to'.**

Reply:We will simplify the presentation.

**Page 2, lines 18-27: This sections is unclear and difficult to understand, probably because (i) the sentences are too long and (ii) the ideas developed in this paragraph are not well organized (e.g in the same sentence (starting line 20 and ending line 24, both the influence on preferential flow of initially wet and initially dry soils are discussed, but it is difficult to understand exactly which arguments refer to which situation)**

Reply: We will re-organize the presentations with help of a native English speaker.

**Page 4 line 16-17: "rainfall events were divides into ones. . ... rains". I was not able to understand this sentence.**

Reply: We will rewrite the sentence to make it clearer. Generally, this sentence means that in this study, duration of a rainfall event is not longer than 24hrs. But in sometimes, the rain continues for a long time and cannot be cut apart by the 24-hr bar, so duration of the rainfall event will last for a longer time, but not longer than 48hrs.

**Page 8, Line 1-2 I was not able to understand this sentence.**

Reply: We will rewrite this sentence.

---

## Author Comment (AC4) · 27 Sep 2016

**General Comments**

**The study of "Impacts of rainfall features and antecedent soil moisture on occurrence of preferential flow: A study at hillslopes using high frequency monitoring" is of considerable significance to the scientific community interested in better understanding the onset of preferential flow. The vast amount of observational data collected by the authors is impressive. However, a significant revision needs to be done before the manuscript becomes suitable for publication.**

**One of the main issues with the manuscript is that it fails to engage the reader as to the importance of this study and how it differs from previous studies. The authors should highlight it in the Introduction section. Secondly, having a separate Discussion section is not the best approach here. Having a Results and Discussion section merged together will give the readers a better understanding of the similarities and disparities between the current study and the previous studies.**

**There are also a lot of grammatical mistakes in the manuscript, which makes the manuscript harder to understand and keep the focus on the actual research part. I suggest the authors have a native English speaking person review the manuscript to improve its overall quality.**

Reply: Thanks a lot for your comments.

We will illustrate more to fill the gap between previous studies and our objectives. Basically, this study was initiated from two considerations. (1) It would be helpful in understanding the processes of subsurface hydrology, if we get the key factors that control the occurrence of preferential flow. Lots of studies have been carried out on this topic. However, contradictory results were obtained in different cases, e.g., the cases of Wu et al. (2014) and Hardie et al. (2013). And to our knowledge, no study on this topic has been carried out in northern China with sub-humid climate and poorly developed underlying soil. Hence, we think this study could be a complementary to the understanding of controlling factors of preferential flow; meanwhile, it would be helpful in understanding hydrological processes of the study area. (2) By far, there are many methods for the detection of preferential flow, but in-suit method is rather limited. The method using wetting front as an indicator, which was proposed by Lin and Zhou (2008) and later improved by Hardie et al. (2013), could be an alternative option. Since this method has been on applied in only two or three cases to our knowledge, it would be of interest to apply it in our study area, where climate and surface condition are different from previous cases. From this point of view, this study could be regard as a case study. And in the revised introduction, we will emphasis more about the difference between the study area and those of the previous studies, so as to illustrate the necessity to conduct studies at this area.

We've been hesitating whether to put Section 4 as discussion or as a part of results at the beginning. Thanks for your suggestion. We will combine Section 3 and Section 4 together as a section of "Results and Discussion". We will make more comparisons between our results with those of the previous studies, so as to make our results sounder and more meaningful.

Concerning on the language issue, we will have a native English speaker for some help during revision. We will check the syntax errors through the manuscript, and improve the presentations.

**Specific Comments**

**The title of paper can be shortened. Consider leaving out the second part of the title, as it does not provide much additional information.**

Reply: Thanks a lot for your suggestion. Given that this study is kind of a case study, we think it would be necessary to present some key information of the study area in the title. We would change the tile into "*impact of rainfall features and antecedent soil moisture on occurrence of preferential flow in a sub-humid catchment*".

**Authors have used the word "rainfalls" at numerous instances throughout the manuscript. It should be "rainfall".**

Reply: We will check through the manuscript to make sure the word is properly used.

**Missing citations in the Reference section. For example, Vogel et al. 2010; Niu, 2003. Please re-check and make sure all the references listed.**

Reply: We will check through the manuscript to make sure the references are properly presented.

**Page 1, Line 14: Remove "," after intensity.**

Reply: the comma will be dropped.

**Page 1, Line 22: Replace "knowledge" with "finding". Using the word "finding" instead of "knowledge" tells the reader the importance of the study.**

Reply: Thanks for your reminding. We will change "knowledge" into "finding".

**Page 1, Line 26: Indent the paragraph.**

Reply: We will re-format the manuscript according to the guidance of the journal.

**Page 2, Line 4: Rewrite 1st sentence. "Among the many" what? Cite the studies after you have addressed what are "among the many".**

Reply: "Many" refers to the factors that influence the occurrence of preferential flow. We will rewrite sentence to make it clearer.

**Page 2, Line 6: Do not begin the sentence with "And". Reword the sentence.**

Reply: Thanks for reminding. We will check through the manuscript.

**Page, Line 20: A 2008 study, in my opinion is not "recent". Rewrite the sentence and try not to refer a 2008 study as "recent" or cite another relevant "recent" study.**

Reply: Thanks for reminding.

**Page 2, Line 20-24: Try to break the sentence "On one hand . . .. . ." It is too long.**

Reply: We are sorry that sentences in the paragraph are long hard to read. We will re-organize this section.

**Page 2, Line 20: swap "became" with "becomes".**

Reply: We will change "became" into "becomes".

**Page 2, Line 24: rewrite the sentence "However, since larger . . .. . .". It is poorly worded and is difficult to understand.**

Reply:We will re-organize this paragraph.

**Page 2, Line 28: "Among the numerous approaches" has only reference. Please cite additional studies if "numerous approaches" are mentioned.**

Reply: We will specify the approaches in revision, by referring to the review work by Allaire et al. (2009) and other studies.

**Page 2, Line 29: Use a semicolon instead of a comma after 1996 to cite two different papers.**

Reply: We will re-format the manuscript according to the guidance of the journal.

**Page 2, Line 3: In the last paragraph of the Introduction, the authors should not only mention what they are doing in the study but why and also highlight how their study is different from previous studies.**

Reply: Thanks for your suggestion. We will illustrate more to fill the gap between previous studies and our objectives. Since this is a case study, we would emphasis more about the difference between this study area with those of previous studies.

**Page 2, Line 8: Indent the first paragraph.**

Reply: We will re-format the manuscript according to the guidance of the journal.

**Page 2, Line 16: Indent the first paragraph. I have noticed this error through out the manuscript. Please fix this.**

Reply: Thanks for reminding. We will re-format the manuscript according to the guidance of the journal.

**Page 3, Line 20: The authors have not mentioned the reason as to why the probes were buried at different depths at different sites. Was it due to varying soil depths at these locations? What was the need to bury the probes further than 60cm if the authors only use 0-60cm for their analysis shown in the Figures?**

Reply: The probes were buried at different depths for other purposes, and only the data of the 0-60cm is used in this study, in order to compare among the sites. We will clarify this issue in revision.

**Page 4, Line 11: Which "data" are the authors referring to?**

Reply: "Data" here refers to Table 2. We will re-organize the presentation to make it clearer.

**Page 2, Line 27: Swap "was" with "is".**

Reply: We will change "was" into "is".

**Page 4, Line 25: "Producer" is not an appropriate word here. May be the authors can use "manufacturer".**

Reply: Thanks a lot for your suggestion. We will use the word "manufacture" instead.

**Page 4, Line 27: ". . . a bunch of studies". Use another way of describing like " a lot of studies" or "other studies".**

Reply: Thanks for reminding. We'll be careful in using the phrase "a bunch of".

**Page 5, Line 21: change ". . .. as was shown" to "as is shown".**

Reply: We will change "was" into "is".

**Page 6, Line 4: Wrong usage of word "respectively".**

Reply: The word "respectively" will be dropped.

**Page 8, Line 22: What does n=233 refer to?**
Reply: "n" refers to the number of samples used for correlation analysis, specifically, it is the cumulative number of the rainfall events measured by the 9 rain gauges and used in analysis.

---

## Author Comment (AC5) · 27 Sep 2016

**This paper tried to evaluate the impact of rainfall features and antecedent soil moisture on occurrence of preferential flow on slope in north China by interpreting response of soil moisture to rainfall. The result showed that occurring frequency of preferential flow was averagely 40.7. Nevertheless, it is my feeling that the authors did not stress enough the limitation of previous researches and their relations with the major objectives. And I also concern about the method that the authors used to analyze the correlation between rainfall features, antecedent soil moisture and frequency of preferential flow. It is not clear to me that how each rainfall feature was divided into 15 sub-ranges with non-uniform increments (P8L5). In my opinion, the method would determine the fitting curves and R2, and so the results depend on the inclination of the authors. Therefore, the authors should provide more explanation. As a general comment, I think that the paper requires major revision before being published.**

Reply: Thanks a lot for your comments.

We will illustrate more to fill the gap between previous studies and our objectives. Basically, this study was initiated from two considerations. (1) It would be helpful in understanding the processes of subsurface hydrology, if we get the key factors that control the occurrence of preferential flow. Lots of studies have been carried out on this topic. However, contradictory results were obtained in different cases, e.g., the cases of Wu et al. (2014) and Hardie et al. (2013). And to our knowledge, no study on this topic has been carried out in northern China with sub-humid climate and poorly developed underlying soil. Hence, we think this study could be a complementary to the understanding of controlling factors of preferential flow; meanwhile, it would be helpful in understanding hydrological processes of the study area. (2) By far, there are many methods for the detection of preferential flow, but in-suit method is rather limited. The method using wetting front as an indicator, which was proposed by Lin and Zhou (2008) and later improved by Hardie et al. (2013), could be an alternative option. Since this method has been on applied in only two or three cases to our knowledge, it would be of interest to apply it in our study area, where climate and surface condition are different from previous cases.

We agree that our way to set sub-ranges is not absolutely robust. Nevertheless, Given that the values of features are not uniformly distributed, we think it's inappropriate to divide the sub-ranges by uniform increments. If the increment were small, only one or even none of the data points would be contained by the sub-ranges with high values, and statistic of frequencies in these sub-ranges would be meaningless. If the increment were large, only one or two sub-ranges could be obtained at low values, the increase trend of frequency would be less visible, and some key information may be lost. In this circumstance, we think the non-uniform sub-ranges would be a reasonable choice. In order to compare among the features, values of the 4 features are all divided into the same number of sub-ranges. Moreover, ranges of the sub-ranges all increase with values by generally the same rates in all the 4 scenarios of features (there are several exceptions in the scenario of "maximum intensity", and it is caused by the limit of value distribution). Therefore, though there are other ways to set sub-ranges, we think our interpretation of data is reasonable, and we will illustrate more to explain the sub-ranges.

**I have listed in the following a number of issues that should be addressed in this paper before publication:**

**1. "Eighty four groups of soil samples were collected from the profiles at all sites 5 except FH4. "(P4L5). The reason should be explained why FH4 was excluded.**

Reply: Information about the soil properties of FH4 was not deliberately excluded, but because of sample missing during particle analysis. We are going to re-sample the soils at this site and analysis their properties.

**2. What is the accuracy of the rain gauges?**

Reply: According to the instruction manual, accuracy of the rain gauge is $\pm 1.0\%$ at rainfall rate lower than 50mm/hr. This information will be added in the revised manuscript.

**3. The slope gradient and aspect, canopy coverage and elevation of each site are suggested to add to Table1, which will help readers to understand the differences of the sites. And more explanations should be given why the sites S1H1-S1H5 and S2H1-S2H3 were set, which seem very close to each other according to Fig1.**

Reply: Thanks for your suggestion. We will add the suggested information into the manuscript. Generally, sites FH1-FH4 are located at flat areas without canopy cover, while sites S1H1- S1H5 and sites S2H1-S2H3 are located at two typical hillslopes, respectively, and both slopes are covered by canopy, especially when leaves are exuberant.

Sites S1H1- S1H5 are on the same hillslope and are close to each other, thus they should be in similar conditions and could be grouped together. So are sites S2H1-S2H3. In this case, comparison could be carried out in statistic, and some discussions on the influences of slope and surface cover could be carried out, e.g., the discussions in Section 4.2.

**4. What the measurement radius of the probes of TDR? The information is important because only the preference flow occurred in this range could be interpreted by the variation of observed soil moisture.**

Reply: The CS616 TDR is a 2-rod probe. Rod diameter is 3.2 mm, rod space is 32 mm, and rod length is 30cm. According the manual of the product, measurement radius of the probe is 5cm, and measurement accuracy is $\pm$2.5% VWC.

**5. What is theoretical basis that a 0.002 cm3/cm3 threshold was set to quantify the responses of water content to infiltration according to a bunch of studies, given accuracies of the applied TDR probes were 0.025cm3/cm3. Whether did previous studies (Blume et al., 2009; Lin and Zhou, 2008) use the TDR probes of same accuracy?**

Reply: In Lin and Zhou's (2008) study, accuracy of the probe is $\pm$4% and relevant resolution is 0.2%. Accuracy of the probes is about $\pm$3% in Blume's et al. (2009) study and $\pm$2.6% in Hardie's et al. (2013) study. They all suggested 0.2% as the threshold of the responses of water content. While in this study, accuracy of probe is 2.5%, and resolution is 0.1%, thus the 0.2% threshold should be reasonable.

**6. The null hypothesis of the Kolmogorov-Smirnov test is usually defined as that the sample is drawn from the reference distribution (in the one-sample case) or that the samples are drawn from the same distribution (in the two-sample case) (such as in XLSTAT). However, "significant difference between every two distribution was set as the null hypothesis in this paper"(Line102-103). Which software was used to carry out the tests?**

Reply: Thanks a lot for your remanding. But theoretically, it would be also reasonable to state that distributions are the same by rejecting the hypothesis of significant difference. We will conduct the test according to your suggestion, to see if the results will be different.

All calculations of this study are carried out on a MATLAB R2014b compiler.

**7. It is difficult to read Fig4 and I suggest change it to a table.**

Reply: Thanks for your suggestion. We will change the figure into a table.

**8. "Contents of organic matter, clay and silt generally decreased with depth, leading to higher saturated hydraulic conductivity. Detailed information at each depth was not listed in Table 2, but was covered by the ranges."(P4L11-12). What is the sampling depth of the data in Table 2?**

Reply: Sampling depth was limited by the depth of underlying bedrock at each site. It was 60cm at FH1-FH3 and 80cm at other sites, except that was 90cm at S2H1 and S2H3. Samples were obtained by 10cm intervals of depth.

**9. The rainfall amount difference between site FH3 and S2H1 is larger than 180mm from 2014/8/22 to 2014/10/31 (Table3) but the distance seems only about 500m. Is it because the logger at S2H1 failed from 2014/10/14 to 2014/10/31(P5L27)? This should be added as notes of Table3.**

Reply: We would not attribute this difference to the failure of data logger during 2014/10/14 to 2014/10/31, since only 0.5mm of rain was precipitated in this period. Given that the difference at the same slope could be up to about 60cm (S1h1 and S1h2), space heterogeneity of rainfall itself and canopy interception should have played a more prominent role.

**10. Fig1 is not contour map but a DEM map.**

Reply: Thanks for your suggestion. We will change Figure 1 into a contour map.